# Consequences of PDGFRα⁺ fibroblast reduction in adult murine hearts

Jill T Kuwabara[1,2†], Akitoshi Hara[1†], Sumit Bhutada[3], Greg S Gojanovich[1], Jasmine Chen[1,2], Kanani Hokutan[1], Vikram Shettigar[4], Anson Y Lee[1], Lydia P DeAngelo[1], Jack R Heckl[1,2], Julia R Jahansooz[1], Dillon K Tacdol[1], Mark T Ziolo[4], Suneel S Apte[3], Michelle D Tallquist[1]*

[1]Center for Cardiovascular Research, John A. Burns School of Medicine, University of Hawaii at Manoa, Honolulu, United States; [2]Department of Cell and Molecular Biology, John A. Burns School of Medicine, University of Hawaii at Manoa, Honolulu, United States; [3]Department of Biomedical Engineering, Cleveland Clinic Lerner Research Institute, Cleveland, United States; [4]Dorothy M. Davis Heart and Lung Research Institute, Department of Physiology and Cell Biology, The Ohio State University Wexner Medical Center, Columbus, United States

**Abstract** Fibroblasts produce the majority of collagen in the heart and are thought to regulate extracellular matrix (ECM) turnover. Although fibrosis accompanies many cardiac pathologies and is generally deleterious, the role of fibroblasts in maintaining the basal ECM network and in fibrosis in vivo is poorly understood. We genetically ablated fibroblasts in mice to evaluate the impact on homeostasis of adult ECM and cardiac function after injury. Fibroblast-ablated mice demonstrated a substantive reduction in cardiac fibroblasts, but fibrillar collagen and the ECM proteome were not overtly altered when evaluated by quantitative mass spectrometry and N-terminomics. However, the distribution and quantity of collagen VI, microfibrillar collagen that forms an open network with the basement membrane, was reduced. In fibroblast-ablated mice, cardiac function was better preserved following angiotensin II/phenylephrine (AngII/PE)-induced fibrosis and myocardial infarction (MI). Analysis of cardiomyocyte function demonstrated altered sarcomere shortening and slowed calcium decline in both uninjured and AngII/PE-infused fibroblast-ablated mice. After MI, the residual resident fibroblasts responded to injury, albeit with reduced proliferation and numbers immediately after injury. These results indicate that the adult mouse heart tolerates a significant degree of fibroblast loss with a potentially beneficial impact on cardiac function after injury. The cardioprotective effect of controlled fibroblast reduction may have therapeutic value in heart disease.

*For correspondence:
michelle.tallquist@hawaii.edu

†These authors contributed equally to this work

Competing interest: The authors declare that no competing interests exist.

## Editor's evaluation

This study is directed at cardiac fibrotic change, which is widespread both under conditions of both normal development and pathological responses, and it is associated with both pump failure and arrhythmic change. A new genetic platform is introduced, in which such processes are reduced. What emerges is a reduction in ventricular, atrial and septal fibroblast density. This study is extremely well done, rigorous, and offers valuable insight for investigators interested in fibrosis, cardiac fibroblast biology, and mechanisms of extracellular matrix remodeling.

## Introduction

Fibroblasts are the primary source of extracellular matrix (ECM) in the heart (*Eghbali et al., 1988*; *Eghbali et al., 1989*). The main physiological role of the cardiac fibroblast is to maintain the ECM by balancing the deposition and degradation of structural ECM proteins and non-structural matricellular proteins (*Tallquist and Molkentin, 2017*). This ECM network serves as a scaffolding that mechanically supports cardiomyocytes, transmits force, and regulates mechanical signals (*Borg and Caulfield, 1979*; *Weber, 1989*). It is thought that ECM-cytoskeletal connections are essential for proper stability and contraction of the cardiomyocyte, indeed, disruptions in these interactions underlie a wide range of cardiomyopathies (*Harvey and Leinwand, 2011*). In addition to matrix production, fibroblasts can functionally couple with cardiomyocytes via connexins (*Ongstad and Kohl, 2016*; *Camelliti et al., 2004*). Although the ECM network is thought to be primarily synthesized and organized by cardiac fibroblasts and integral to proper cardiomyocyte function (*Eghbali et al., 1988*; *Eghbali et al., 1989*; *Ieda et al., 2009*), there is limited data on homeostatic, fibroblast-specific ECM production in vivo and the potential impact of altering fibroblast numbers in the adult heart.

In response to cardiac injury, fibroblasts rapidly adopt an activated phenotype resulting in increased proliferation and deposition of a collagen-rich ECM (*Ivey et al., 2018*; *Christia et al., 2013*; *Moore-Morris et al., 2014*; *Fu et al., 2018*). An initial adaptive response maintains structural integrity of the damaged myocardium and prevents rupture (*Cleutjens et al., 1995*; *Factor et al., 1987*). However, prolonged and excessive ECM deposition impairs cardiac compliance due to ventricular wall stiffening (*Talman and Ruskoaho, 2016*; *Frangogiannis, 2019*) and can also disrupt electrical transmission between cardiomyocytes leading to contractile dysfunction (*Beltrami et al., 1994*). Despite its clinical and pathophysiological significance, no interventions currently exist to mitigate or reverse cardiac fibrosis (*Gourdie et al., 2016*). The cardiac fibroblast has emerged as an ideal candidate to regulate fibrosis that accompanies cardiac injury, but its role remains relatively obscure.

Recent studies have focused on genetically targeting myofibroblasts or activated fibroblasts as a potential therapy for heart disease because of their contribution to fibrotic scar formation and subsequently reduced heart function (*Moore-Morris et al., 2014*; *Kanisicak et al., 2016*; *Khalil et al., 2019*; *Molkentin et al., 2017*; *Kaur et al., 2016*). When chimeric antigen receptor-T cells were used to specifically target an endogenous cell-surface glycoprotein on activated fibroblasts, fibrosis was reduced, leading to better cardiac function (*Aghajanian et al., 2019*). Similarly, ablation of activated fibroblasts (*Kaur et al., 2016*) or activated fibroblast-specific depletion of *Grk2*, a downstream effector of G protein-coupled receptors known to be elevated in patients with heart failure (*Travers et al., 2017*), was shown to be cardioprotective following injury. While these studies suggest beneficial outcomes from fibroblast reduction, others have observed increased lethality and ventricular wall rupture, presumably due to reduced ECM deposition (*Kanisicak et al., 2016*; *Molkentin et al., 2017*). When *Hsp47* (*Khalil et al., 2019*), *Fstl1* (*Kretzschmar et al., 2018*; *Maruyama et al., 2016*), or *Smad3* (*Kong et al., 2018*) were disrupted in activated fibroblasts, decreased cardiac function and rupture were observed. Taken together, these data suggest that manipulation of fibroblast numbers and matrix deposition may require a more nuanced approach based on improved fundamental knowledge. Considering that anti-fibrotic therapies are currently being proposed as treatments for heart pathologies, a thorough evaluation and understanding of fibroblast activities during tissue homeostasis is necessary but is presently unavailable.

Given the lack of data specifically addressing fibroblast functions in the uninjured heart, we reduced fibroblast numbers in the adult murine heart using an inducible fibroblast-specific Cre line. Surprisingly, no mortality resulted, even though fibroblast loss was sustained up to 7 months post-induction. Moreover, heart function, protein composition, and proteolytic turnover were largely well-maintained despite the level of fibroblast depletion. Analysis of matrix components, such as type I, IV, and VI collagen, demonstrated that the type I collagen fibrillar network appeared relatively unchanged, whereas changes in microfibrillar collagen and basement membrane components were observed. Fibroblast-ablated hearts also showed improved function after myocardial infarction (MI) and angiotensin II/phenylephrine (AngII/PE) infusion, suggesting that an underlying reduction in fibroblast numbers or cardiac adaptation to fewer fibroblasts may be protective. Our findings reveal that fibroblast reduction before injury does not negatively affect the heart and may provide beneficial effects when cardiac remodeling or repair are provoked by injury.

# Results

## Genetic ablation of fibroblasts during cardiac homeostasis

After heart injury, fibroblasts are essential for the generation of replacement fibrous ECM, and multiple studies have suggested that disruption of fibroblast expansion may lead to wall rupture (*Kanisicak et al., 2016*; *Molkentin et al., 2017*). Although it is thought that fibroblasts are indispensable for maintaining ECM structure during cardiac homeostasis, we recently demonstrated that a loss of up to 50% of resident fibroblasts was inconsequential for the architecture and function of the heart (*Ivey et al., 2019*). These results led us to investigate whether the heart can tolerate a further reduction in fibroblast numbers. As PDGFRα is expressed in a majority of murine adult cardiac fibroblasts (*Ivey et al., 2018*; *Pinto et al., 2016*), we induced diphtheria toxin A (DTA) expression using a *Pdgfra-CreERT2/+* mouse line to deplete resident fibroblasts (*Chung et al., 2018*). The DTA expression following tamoxifen treatment led to fibroblast loss within several days. We refer to these mice as fibroblast-ablated or ablated. Cre activity, indicated by a tdTomato reporter, demonstrated that a reduction in PDGFRα lineage cells occurred by 14 days post-induction (*Figure 1A and B*). To investigate fibroblast reduction by an orthogonal approach, we assayed for vimentin expression (*Ali et al., 2014*), which showed fewer vimentin-positive cells in ablated hearts compared to controls (*Figure 1A*). Furthermore, we determined if other cell populations activated collagen production in compensation using a transgenic collagen reporter, collagen1a1-green fluorescent protein (GFP) (*Col1a1*-GFP). This reporter expresses GFP under the control of the *Col1a1* promoter (*Moore-Morris et al., 2014*; *Pinto et al., 2016*; *Acharya et al., 2012*; *Smith et al., 2011*; *Yata et al., 2003*). An overall reduction in collagen-expressing cells was observed in the ablated mice (*Figure 1A and C–E*), suggesting that PDGFRα-expressing fibroblasts remain the primary producers of type I collagen even after their numbers are decreased. Reduced reporter activity was not limited to the ventricles, as reductions were also observed in the atria and septum, while aortic expression of the GFP reporter remained relatively constant (*Figure 1C*). The reduced level of cells with type I collagen promoter activity persisted 7–15 months after ablation (*Figure 1E*), suggesting compensation for fibroblast loss does not occur by the expansion of the remaining PDGFRα or other cell populations for up to one year after ablation.

Flow cytometry confirmed a reduction in PDGFRα+ fibroblast numbers while the number of endothelial (CD31+) and myeloid cells (CD11b+) were unaffected (*Figure 1F*). It should be noted that the mean fluorescence intensity of surface PDGFRα decreased in mice heterozygous for *Pdgfra* (*Figure 1—figure supplement 1*). Western blot, immunostaining, and qPCR confirmed a decrease in PDGFRα protein and transcript in ablated hearts 2–7 months post-induction (*Figure 1G–I* and *Figure 1—figure supplement 2A*). Additionally, other fibroblast-related genes, including those that define alternative fibroblast populations identified in single-cell RNAseq studies (*Skelly et al., 2018*; *Farbehi et al., 2019*) were also reduced (*Figure 1I*). As PDGFRβ expressing cells are suggested to adopt fibroblast characteristics after injury (*Peisker et al., 2022*; *Henderson et al., 2013*), we determined if the remaining collagen expressing cells were derived from a PDGFRβ population. We found that in ablated hearts PDGFRβ+ cells maintained a vascular distribution similar to control hearts (*Figure 1—figure supplement 3*) and that there was no significant increase in PDGFRβ+ cells in the *Col1a1*-GFP+ population (*Supplementary file 1a*). Finally, gene expression of primary fibroblast cultures selected by adhesion enrichment demonstrated that the adherent cells from ablated hearts did not upregulate typical fibroblast genes and in some cases had a reduced expression of fibroblast-related genes (*Figure 1J*). These data suggest that little compensation occurs when PDGFRα expressing fibroblasts are ablated. Therefore, genetic fibroblast ablation is efficient, effectively reduces the resident fibroblast population within 14 days, and lasts up to 1 year post-induction in the adult murine heart.

Despite a significant reduction in fibroblasts, ablated hearts were histologically and functionally indistinguishable from controls. Although body weight was reduced in long-term fibroblast-ablated mice, the heart weight to body weight ratio was proportionate in aged females and increased slightly in aged males (*Figure 1—figure supplement 2B-D*). Cardiomyocyte cross-sectional area (CSA) determined by wheat germ agglutinin (WGA) staining was similar in control and ablated hearts (*Figure 1—figure supplement 2E*). Echocardiography and pressure-volume loop analyses revealed that control and ablated hearts maintained a similar ejection fraction, left ventricular (LV) chamber size, diastolic function, LV filling pressure, and rate of systolic and diastolic blood pressure change up to 7 months following fibroblast ablation (*Figure 1—figure supplement 2F-K*). Because we induced DTA in all PDGFRα-expressing cells, we used a basic metabolic panel to investigate any overt blood chemistry

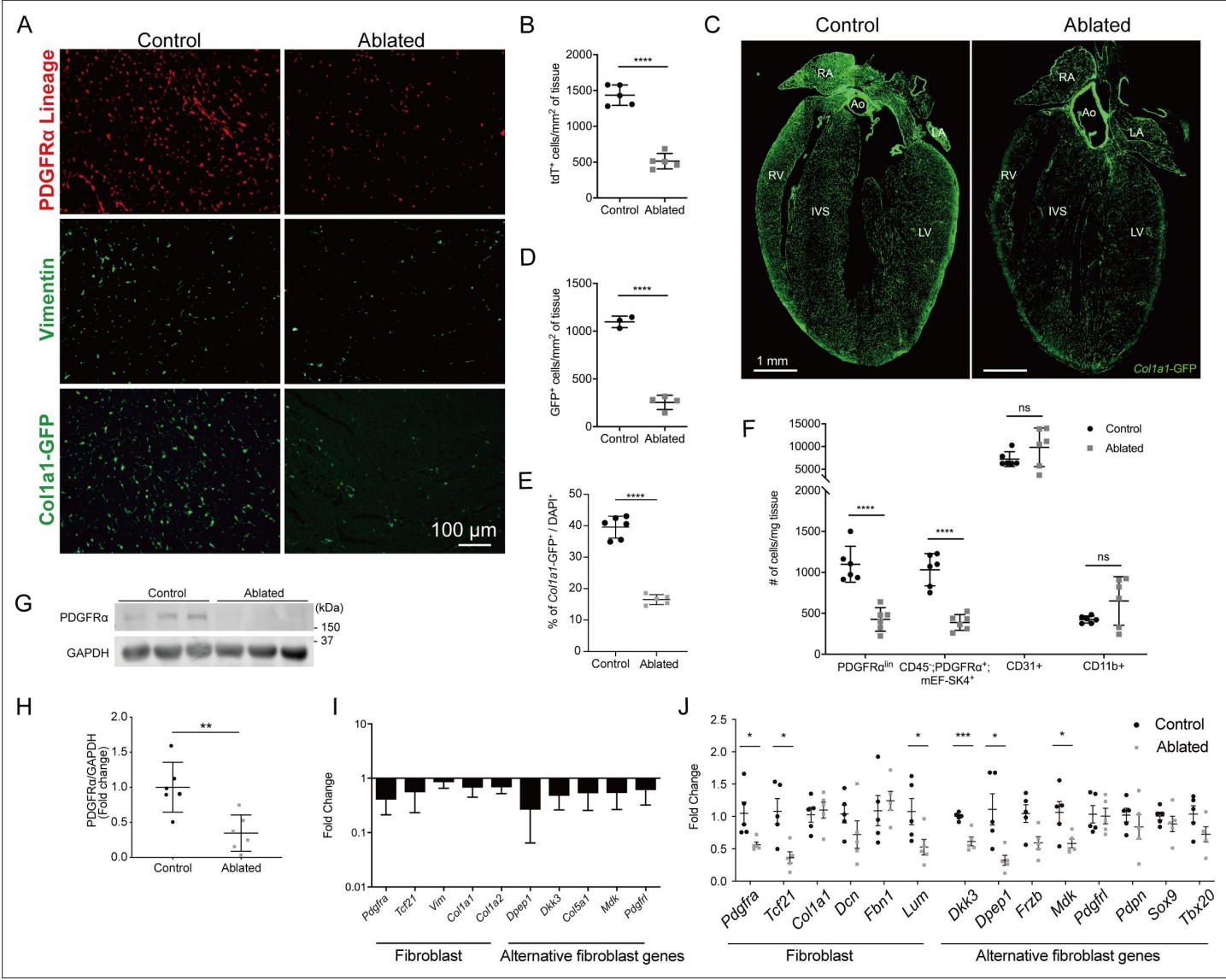

**Figure 1.** Loss of fibroblasts in adult ablated hearts. (**A**) Representative images of PDGFRα lineage (tdTomato), vimentin immunostaining, and *Col1a1*-GFP reporter fluorescence in the left ventricular (LV) myocardium. A representative from a minimum of three biological replicates for each staining. (**B**) Quantification of tdTomato. Three fields of view (FOV) at 20× from two non-consecutive sections per biological replicate were quantified. Control: n=5; ablated: n=5. (**C**) A four chamber view of *Col1a1*-GFP expression. Ao: aorta; IVS: interventricular septum; LA: left atrium; LV: left ventricle; RA: right atrium; RV: right ventricle. A representative from a minimum of three biological replicates. (**D–E**) Quantification of *Col1a1*-GFP (**D**) 2 weeks after induction (control: n=3; ablated: n=4) and (**E**) over 1 year after induction (control n=6; ablated: n=5). (**F**) Quantification of fibroblasts (PDGFRα^lin, CD45^-; PDGFRα^+; mEF-SK4^+), endothelial cells (CD31^+), and immune cell (CD11b^+) populations by flow cytometry. Neutrophils, mast cells, and eosinophils were fewer than 100 cells/mg of tissue in both control and ablated hearts. Control: n=6; ablated: n=6. (**G, H**) Representative western blot of whole ventricle lysates and band intensities. Control: n=6; ablated: n=6. (**I**) The quantitative real-time polymerase chain reaction (qRT-PCR) analysis of selected fibroblast and alternative fibroblast genes in whole ventricles from ablated hearts compared to controls. Control: n=3; ablated: n=4. (**J**) The qPCR analysis of selected fibroblast and alternative fibroblast genes in primary fibroblasts isolated from indicated genotypes 8 months after induction. (**A–D, F**) 4–14 days after induction. Control: n=5; ablated: n=5. (**C, I**) 8–14 months after induction. (**G, H**) 2–4 months after induction. Results are mean ± SD. Statistical significance was determined by an unpaired t-test. ns: not significant, p>0.05; *p≤0.05; **p≤0.01; ***p≤0.001; ****p≤0.0001.

The online version of this article includes the following source data and figure supplement(s) for figure 1:

**Source data 1.** Full unedited western blots.

**Figure supplement 1.** The PDGFRα detection by flow cytometry.

**Figure supplement 2.** Basal phenotype after short- and long-term fibroblast loss.

**Figure supplement 3.** The PDGFRβ expressing cells.

abnormalities. The only observed abnormality was a lower hematocrit suggesting that ablated animals were moderately anemic (*Supplementary file 1b*). Taken together, these data signify that fibroblast loss is well-tolerated in the heart in the absence of injury.

## Collagen fiber organization and basement membrane protein distribution

Because fibroblasts are responsible for the majority of type I collagen production in the heart (*Eghbali et al., 1988*; *Eghbali et al., 1989*), we evaluated collagen levels by immunostaining and hydroxyproline content and found that type I collagen and hydroxyproline quantity remained constant in ablated hearts (*Figure 2A, E, I*). To examine the ultrastructure of the ECM, we visualized decellularized LV tissue by scanning electron microscopy (SEM). The endomysial weaves of collagen fibrils surrounding individual muscle fibers were less dense and thinner 2 months post-fibroblast ablation (*Figure 2J*). These results indicate that after fibroblast ablation, the adult murine heart retained physiological collagen abundance with modestly affected fibrillar collagen organization.

Based on our previous findings, a 50% reduction in cardiac fibroblasts resulted in subtle alterations in laminin patterning (*Ivey et al., 2019*). Therefore, we hypothesized that this phenotype would be exacerbated with greater fibroblast loss. At 1 month post-induction, control and fibroblast-ablated hearts had similar distribution and staining intensity of basement membrane proteins, including laminin, collagen IV, and collagen VI (*Figure 2B–D*). However, at 7 months post-induction, reductions in laminin, collagen IV, and collagen VI were observed by immunostaining in fibroblast-ablated hearts (*Figure 2F–H and K*). By western blot analysis, we observed a reduction in collagen VI but not laminin and collagen IV (*Figure 2L–M*), demonstrating that the distribution of some basement membrane proteins, rather than extractable total protein, was affected by fibroblast loss. These data suggest that loss of fibroblasts may disrupt basement membrane composition and proximate networks.

## Proteomics and degradomics of the ECM in fibroblast-ablated hearts

Fibroblasts are key in generating and remodeling many ECM components (*Camelliti et al., 2005*). Therefore, we evaluated cardiac protein composition by label-free quantitative shotgun proteomics using liquid chromatography-tandem mass spectrometry (LC-MS/MS)-based analysis of the decellularized ventricle isolated from control and fibroblast-ablated mice. The elicited cardiac proteomes showed differences in protein composition (*Figure 3A* and *Figure 3—figure supplement 1A*), as demonstrated by principal component analysis (PCA) and unsupervised hierarchical clustering showed distinct segregation of these decellularized proteomes (*Figure 3—figure supplement 1B-C*). A statistical analysis of these proteomes identified several significantly altered ECM proteins, including collagen VI α6 chain, collagen VIII α1 chain, chondroitin sulfate proteoglycan 4 (NG-2), and dermatopontin (*Figure 3B* and *Figure 3—figure supplement 1C*). The top affected pathways defined by Ingenuity Pathway Analysis (IPA) were GP6 signaling, apelin liver signaling, and the intrinsic and extrinsic prothrombin activation pathways (*Figure 3—figure supplement 1D*). Network analysis of all identified ECM protein ratios in IPA identified a molecular network associated with tissue development, cardiovascular system development and function, and cell morphology (*Figure 3C*). We further investigated overall protease activity by gelatin zymography. A clear reduction in matrix metalloproteinase (MMP) activity was observed in ablated hearts, suggesting that loss of fibroblasts not only leads to moderately reduced matrix production but also a distinct reduction in matrix degradation which may preserve the existing ECM (*Figure 3D*).

To further delineate any proteolytic changes between control and fibroblast-ablated hearts, we used a specialized mass spectrometry-based approach called terminal amine isotopic labeling of substrates (TAILS; *Martin et al., 2020*). TAILS is an N-terminomics method that specifically identifies proteolytically cleaved (internal) peptides by labeling free N-termini at the protein level prior to trypsin digestion and LC-MS/MS (*Kleifeld et al., 2010*). These free N-termini are a result of either proteolytic cleavage by proteases or proteins undergoing natural processes such as alternate splicing or removal of signal/pro/trans peptide (*Figure 3—figure supplement 2A*). Therefore, to distinguish between naturally occurring processes and proteolytically cleaved peptides, further bioinformatics tools were applied (*Fortelny et al., 2015*) to determine the position of the identified peptide within the protein and if the peptide matched with the known cleavage site for these natural processes they were considered as natural peptides and were ruled out as proteolytically cleaved peptides. The remaining

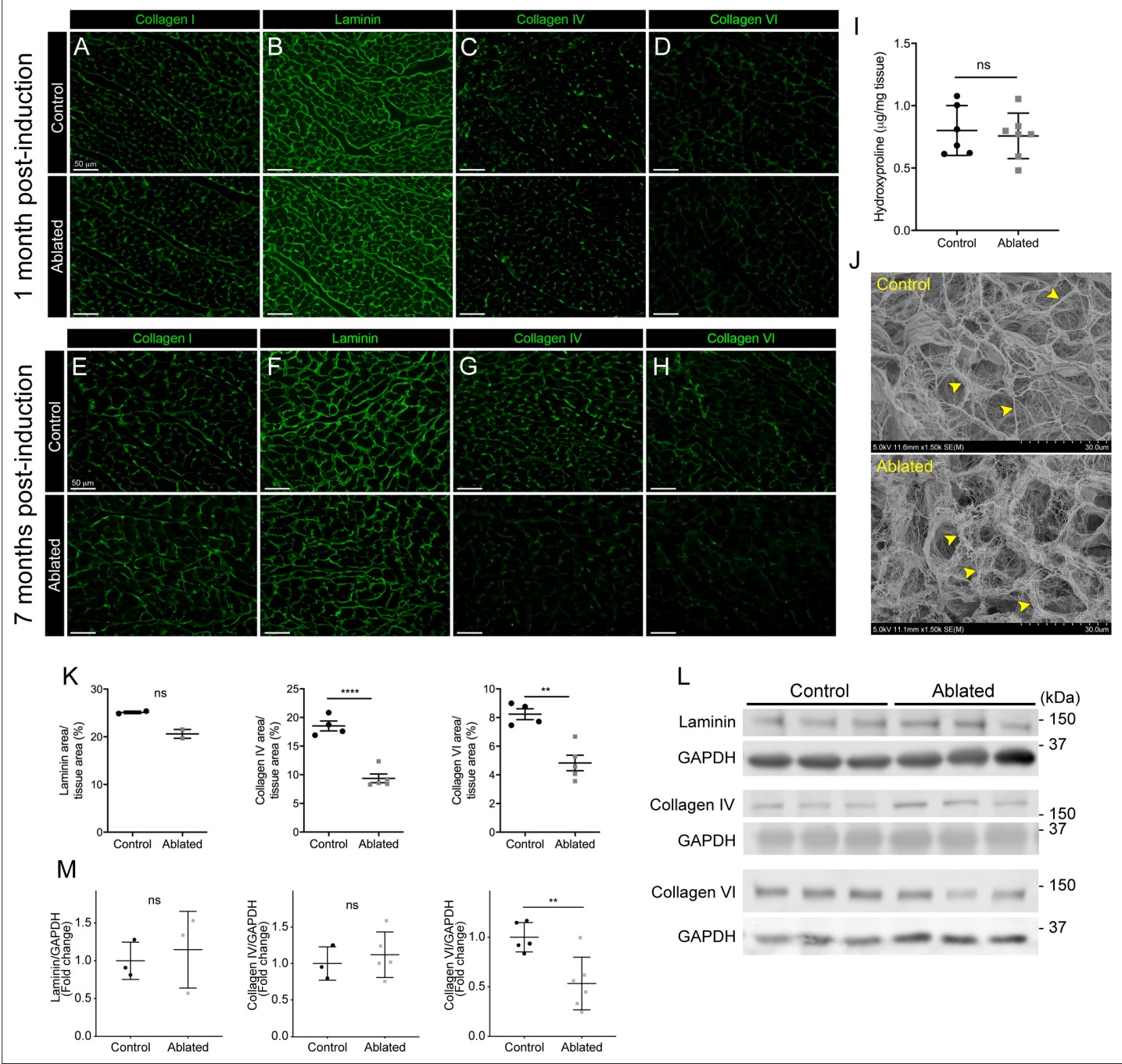

**Figure 2.** Collagen distribution and basement membrane alterations following fibroblast ablation. (**A–H**) Representative images of (**A, E**) collagen I, (**B, F**) laminin, (**C, G**) collagen IV, and (**D, H**) collagen VI immunostaining at indicated post-induction time points. A representative from a minimum of three biological replicates for each staining. (**I**) Hydroxyproline content from whole ventricle tissue 5 months after induction. Control: n=6; ablated: n=7. (**J**) Representative scanning electron microscopy (SEM) images of decellularized left ventricular (LV) tissue at 2 months post-induction. Arrowheads indicate collagen fibrils. Control: n=3; ablated: n=3. (**K**) Area of laminin (control: n=2; ablated: n=2), collagen IV (control: n=4; ablated: n=5), and collagen VI (control: n=4; ablated: n=5) staining normalized to tissue area 7 months after induction. (**L–M**) Western blot analysis of whole ventricle lysate 7 months after induction for laminin (control: n=3; ablated: n=3), collagen IV (control: n=3; ablated: n=5), and collagen VI (control: n=5; ablated: n=6). Results are mean ± SD. Statistical significance was determined by an unpaired t-test or a Mann-Whitney U test. ns: not significant, p>0.05; *p≤0.05; **p≤0.01; ****p≤0.0001.

The online version of this article includes the following source data for figure 2:

**Source data 1.** Full unedited western blots.

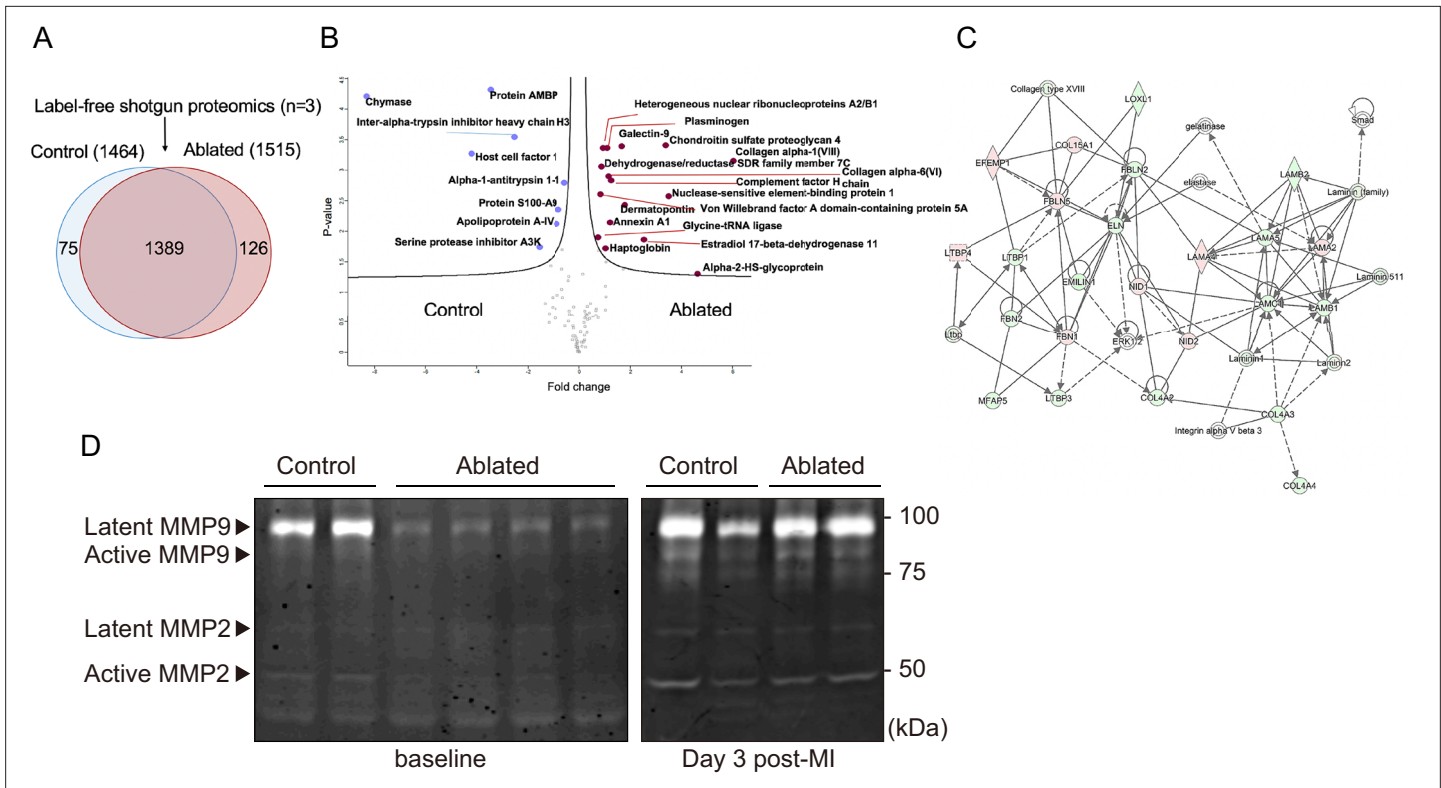

**Figure 3.** Minimal change in matrisome after fibroblast ablation. (**A**) Venn diagram of proteins identified by label-free quantitative shotgun proteomics from decellularized control and ablated heart tissue >2 months after induction. (**B**) Volcano plot showing the distribution of quantified proteins by shotgun proteomics according to p-value and fold change. Blue dots indicate proteins with higher abundance in control ventricles, and red dots indicate proteins with higher abundance in fibroblast-ablated ventricles (all dots are false discovery rate [FDR] <0.05). The p-values are calculated from the data of three replicates. (**C**) Extracellular matrix (ECM) network generated using Ingenuity Pathway Analysis (IPA). Red and green nodes represent up- and down-regulated proteins in control samples, respectively. Empty nodes represent proteins that were not identified in this study but extrapolated from the IPA database. Dashed and continuous lines represent indirect and direct relationships between the proteins, respectively. Control: n=3; ablated: n=3. (**D**) Gelatin zymography of whole ventricles. Control: n=2, 2; ablated: n=4, 2 (baseline and myocardial infarction [MI], respectively).

The online version of this article includes the following source data and figure supplement(s) for figure 3:

**Source data 1.** Full unedited gelatin zymogram.

**Figure supplement 1.** Shotgun proteomics of decellularized heart tissue.

**Figure supplement 2.** N-terminomics of whole ventricle tissue.

**Figure supplement 3.** Potential proteolytically cleaved peptides mapped on myosin-6.

peptides were further filtered and considered only if they had an isobaric tag at the N-terminus as these free N-terminus are most likely generated by proteolytic cleavages and not by trypsin digestion. Statistical tests were applied to determine quantitative differences based on the isobaric tag ratios between the two cohorts. The 8-plex iTRAQ TAILS analysis identified a total of 1834 blocked/labeled N-termini, of which 1519 were internal N-termini (*Figure 3—figure supplement 2B-C*). Of these 1519 internal peptides, 62 peptides were differentially abundant and of these, 15 high confidence internal peptides corresponding to 8 proteins were more abundant in fibroblast-ablated hearts as compared to controls, implying higher proteolytic modification of these proteins, whereas 54 internal peptides corresponding to 30 proteins had higher abundance in controls (*Supplementary file 1c and d*), thus, reconfirming that ablation of fibroblasts leads to less proteolytic activity. Myosin-6 displayed the most internal peptide difference between control and ablated hearts (*Figure 3—figure supplement 3*). Several proteolytic cleavages were detected in other cellular proteins, especially actin isoforms, but no ECM-derived internal peptides were identified as statistically significant. Indeed, pathway analysis of the TAILS peptides identified an impact on cellular metabolism (e.g. mitochondrial dysfunction, oxidative phosphorylation, and sirtuin signaling) and the cytoskeleton and cell-cell adhesion (e.g.

actin cytoskeleton and tight junction signaling), suggesting that cardiomyocyte remodeling may be occurring (*Figure 3—figure supplement 2D*).

## Cardiac responses after AngII/PE infusion

These data demonstrated that fibroblast loss does not lead to obvious structural or compositional deficiencies at baseline. Therefore, we employed a disease model to determine the impact of fibroblast loss on reactive fibrosis using AngII/PE infusion to induce arterial hypertension, adaptive cardiac hypertrophy, and remodeling. Fibroblast levels remained relatively reduced as indicated by tdTomato expression, and fibroblasts activated by AngII/PE infusion demonstrated a perimysial pattern (*Figure 4—figure supplement 1*). Fibroblast ablation did not affect cardiac mass or lung weight (*Figure 4A–B*). Moreover, LV chamber size during diastole was not significantly changed but was slightly reduced during systole (*Figure 4C–D*). While control mice had a modest, sustained reduction in the LV ejection fraction, the ejection fraction of hearts in ablated mice recovered to near baseline levels (*Figure 4E*).

AngII stimulates hypertrophy in cardiomyocytes through angiotensin type 1 receptors (*Sadoshima et al., 1993*). While we did not observe an increase in heart weight to body weight ratio, we found an increase in cardiomyocyte CSA in AngII/PE treated, fibroblast-ablated hearts at both 14 days and 28 days compared to control hearts (*Figure 4F*). We also observed a significant reduction in collagen deposition in the adventitial and interstitial regions of the LV in fibroblast-ablated mice (*Figure 4G–H*). Taken together, these data suggest that ablating fibroblasts before injury results in greater cardiomyocyte hypertrophy and reduced fibrosis leading to normalization of ejection fraction.

## Gene expression profiling of fibroblast-ablated hearts

As suggested earlier, few detectable differences occurred between control and ablated hearts as evidenced by gene expression profiling. Only 10 genes were more highly expressed in ablated hearts, and 34 genes were reduced in baseline ablated hearts compared to controls. Many of these were related to extracellular space and ECM (*Figure 5—figure supplement 1A-B*). To identify potential molecular mechanisms that underlie the cardioprotective aspect of resident fibroblast loss during pathology, we evaluated differentially expressed genes by microarray analysis in fibroblast-ablated hearts compared to controls after 14 days of AngII/PE infusion. Whereas 45 genes were upregulated, and 204 genes were downregulated in ablated hearts (*Figure 5—figure supplement 1A*). Gene ontology (GO) analysis revealed that the downregulated genes were involved in cell adhesion, collagen binding, collagen fibril organization, and ECM organization (*Figure 5A*). The upregulated genes are involved in the cellular response to extracellular stimulus and localize to the membrane, mitochondria, and Z-disc (*Figure 5A*). Consistent with the observed reduction in fibrosis, we found lowered ECM-related mRNAs in ablated hearts by microarray analysis, including *Col1a1*, *Col1a2*, *Col6a2*, *Col6a3*, *Fn1*, *Tnc*, *Nid1*, and *Fbn1* (*Figure 5B*). We also observed a reduction in the expression of genes indicative of fibroblast activation, including *Postn* and *Col3a1* (*Figure 5B–C*), suggesting that ablation of the resident fibroblast population reduces the activated fibroblast population. One characteristic of heart failure due to pathological hypertrophy is the reactivation of the fetal gene program (*Frey and Olson, 2003*). However, *Nppa*, *Nppb*, and *Myh7* mRNA levels were lower in ablated hearts compared to controls after 14 days of AngII/PE infusion (*Figure 5C*), suggesting that pathological hypertrophy may be reduced in ablated hearts. Taken together, these data suggest that decreased fibroblast abundance not only led to reduced ECM gene expression but also elicited a potentially beneficial physiological hypertrophy program in cardiomyocytes.

## Disruption of sarcomere shortening and calcium kinetics in cardiomyocytes

Given the role of cardiomyocyte integrin signaling through interaction with basement membrane components (*Yang et al., 2014*; *Yang et al., 2015*) and the observed changes in basement membrane composition of fibroblast-ablated hearts, we determined whether fibroblast loss led to alterations in cardiomyocyte function. Cardiomyocyte function was measured by recording contraction and calcium transients in individual cardiomyocytes. At baseline, we found that cardiomyocytes from ablated hearts had reduced sarcomeric shortening (*Figure 6A*), increased departure velocity (*Figure 6B*), and slower relaxation (*Figure 6C*). Differences were also observed in sarcomeric shortening and systolic velocity

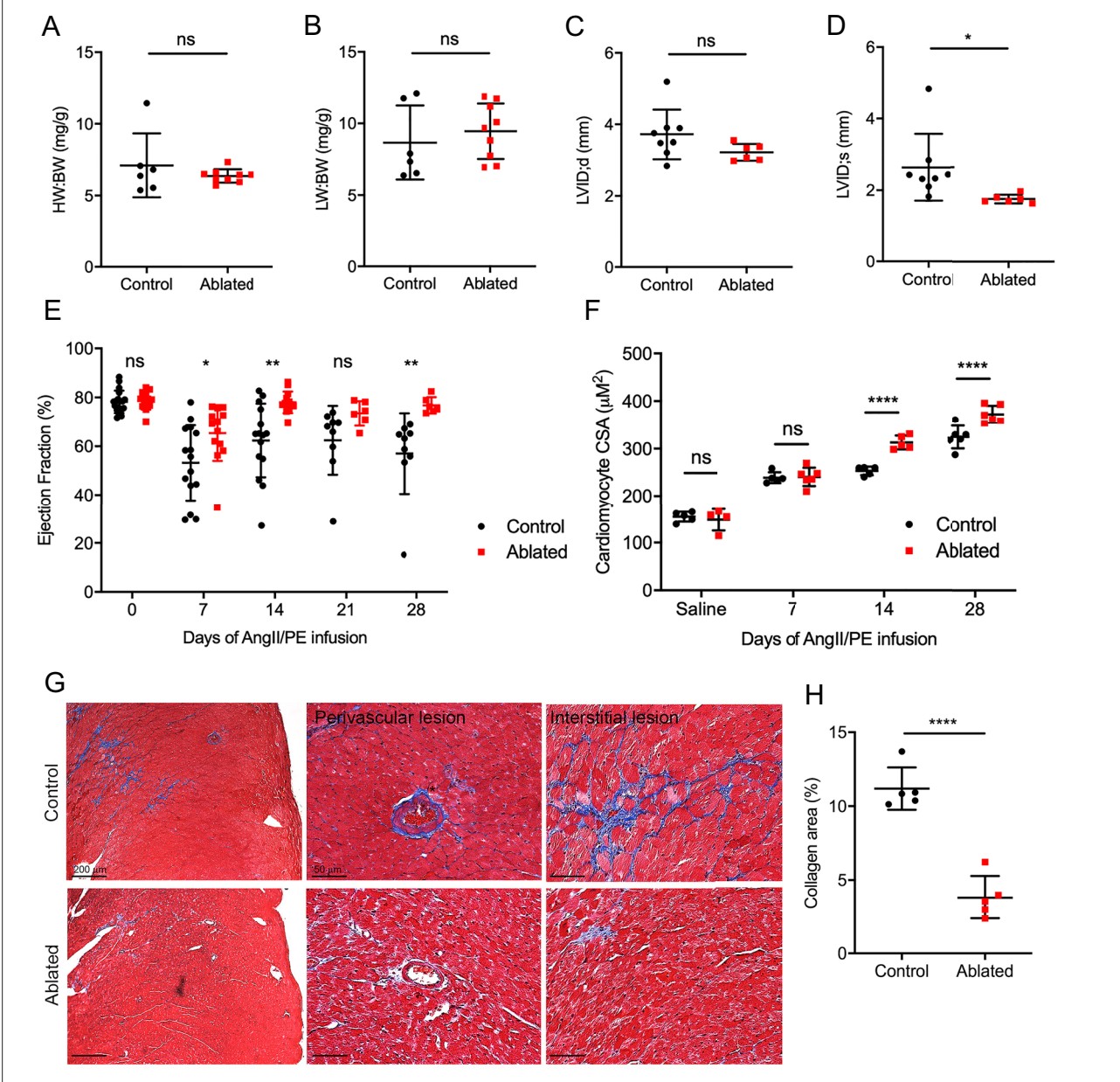

**Figure 4.** Effects of fibroblast loss on heart function and collagen accumulation after angiotensin II and phenylephrine (AngII/PE) infusion. (**A**) Heart weight to body weight (HW:BW) ratio, (**B**) lung weight to body weight (LW:BW) ratio (control: n=6; ablated: n=9), (**C**) diastolic left ventricle internal diameter (LVID), and (**D**) systolic LVID (control: n=8; ablated: n=6) after 28 days of AngII/PE infusion. (**E**) Ejection fraction (EF; control: n=15, 15, 15, 9, 9; ablated: n=13, 13, 13, 6, 6 [baseline, 7, 14, 21, and 28 days AngII/PE, respectively]) and (**F**) cardiomyocyte cross-sectional area (CSA) after AngII/PE infusion (control: n=5, 5, 5, 6; ablated: n=4, 6, 5, 6 [baseline, 7, 14, and 28 days AngII/PE, respectively]). (**G**) Representative images of trichrome staining of the LV myocardium showing perivascular and interstitial regions after 28 days of AngII/PE infusion. (**H**) Quantification of percent collagen in control and ablated hearts. Control: n=5; ablated: n=5. (**A–H**) >7 weeks post-induction at the time of AngII/PE infusion. Results are mean ± SD. The EF results are mean ± SEM. Statistical significance was determined by an unpaired t-test. ns: not significant, p>0.05; *p≤0.05; **p≤0.01; ****p≤0.0001.

The online version of this article includes the following figure supplement(s) for figure 4:

**Figure supplement 1.** Fibroblast expansion after angiotensin II/phenylephrine (AngII/PE) infusion.

after 14 days of AngII/PE infusion (*Figure 6A–B*). Because calcium is a major determinant of cardiac contractility (*Katz and Lorell, 2000*), we examined intracellular calcium kinetics in cardiomyocytes from fibroblast-ablated hearts. Calcium amplitude during contraction was similar between cardiomyocytes from control and ablated hearts at baseline and after AngII/PE infusion (*Figure 6D*). After AngII/PE infusion, calcium amplitude in cardiomyocytes increased from baseline, indicating the expected

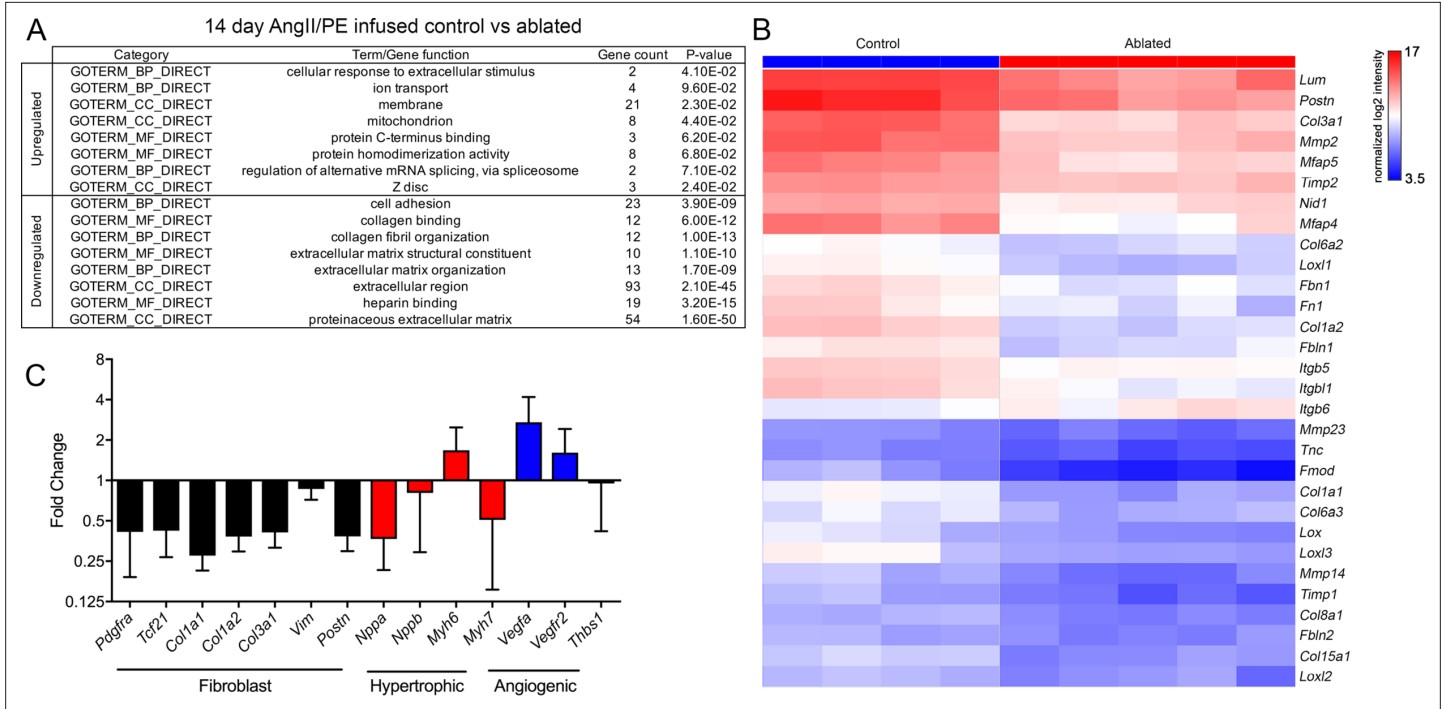

**Figure 5.** Differential gene expression after angiotensin II/phenylephrine (AngII/PE) infusion. (**A**) Gene ontology (GO) analysis of differentially expressed genes between control and ablated hearts infused with AngII/PE for 14 days. All genes included had fold change values of $\log_2 \geq 2$ or $\leq -2$ and $p \leq 0.05$. (**B**) Hierarchical clustering of extracellular matrix (ECM)-related genes differentially expressed in 14-day AngII/PE infused control and ablated. Control: n=4; ablated: n=5. (**C**) The qPCR analysis in whole ventricle tissue infused with 14 days of AngII/PE from ablated hearts compared to controls. The *18s* was used as a housekeeping gene. Control: n=4; ablated: n=5. Results are mean ± SD.

The online version of this article includes the following figure supplement(s) for figure 5:

**Figure supplement 1.** Differentially expressed genes at baseline and after 14 days of angiotensin II/phenylephrine (AngII/PE) infusion.

response to drug infusion (***Figure 6D***). Calcium kinetics, indicated by the decay time constant and the time to reach baseline, was slower at baseline and after AngII/PE infusion in fibroblast-ablated hearts (***Figure 6E–F***). These data suggest that sarcomere shortening and calcium handling are altered in cardiomyocytes after fibroblast reduction. Data from TAILS analysis also supports changes in cardiomyocyte contraction, energy metabolism, and mitochondrial respiration, as internal peptides from proteins involved in these processes were significantly changed in abundance in fibroblast-ablated hearts compared to controls. These included peptides from myosin-6, other myosins, several actin isoforms, and mitochondrial enzymes (***Supplementary file 1c and d***). While these phenomena do not appear to negatively impact heart function, the altered sarcomere contraction and slowed calcium decline in cardiomyocytes may precondition and protect the heart during drug agonist-induced fibrosis.

## Resident fibroblast ablation mitigates cardiac impairment after MI

The fibrotic response to injury is variable and disease-dependent (***Kong et al., 2014***). The MI typically results in cardiomyocyte death and replacement fibrosis, while reactive fibrosis observed with AngII/PE infusion is thought to be induced by cardiac stress and inflammation (***Frangogiannis, 2019***). Depletion of activated fibroblasts after MI leads to reduced collagen production and increased morbidity due to ventricular wall rupture (***Kanisicak et al., 2016***). We next sought to understand how hearts with pre-existing resident fibroblast loss fared after MI. To address this question, we permanently ligated the left anterior descending (LAD) artery >7 weeks post- induction. At 10 weeks post-MI, ~77% of control mice (7/9) and 100% of ablated mice (7/7) survived. Compared to controls, fibroblast ablation did not affect cardiac mass or lung weight, which are key parameters of left-sided heart failure (***Figure 7A–B***). Echocardiography revealed that LV chamber sizes during diastole and systole were comparable at 4 weeks post-MI, but LV chamber sizes increased only in the control mice 4–10 weeks after MI

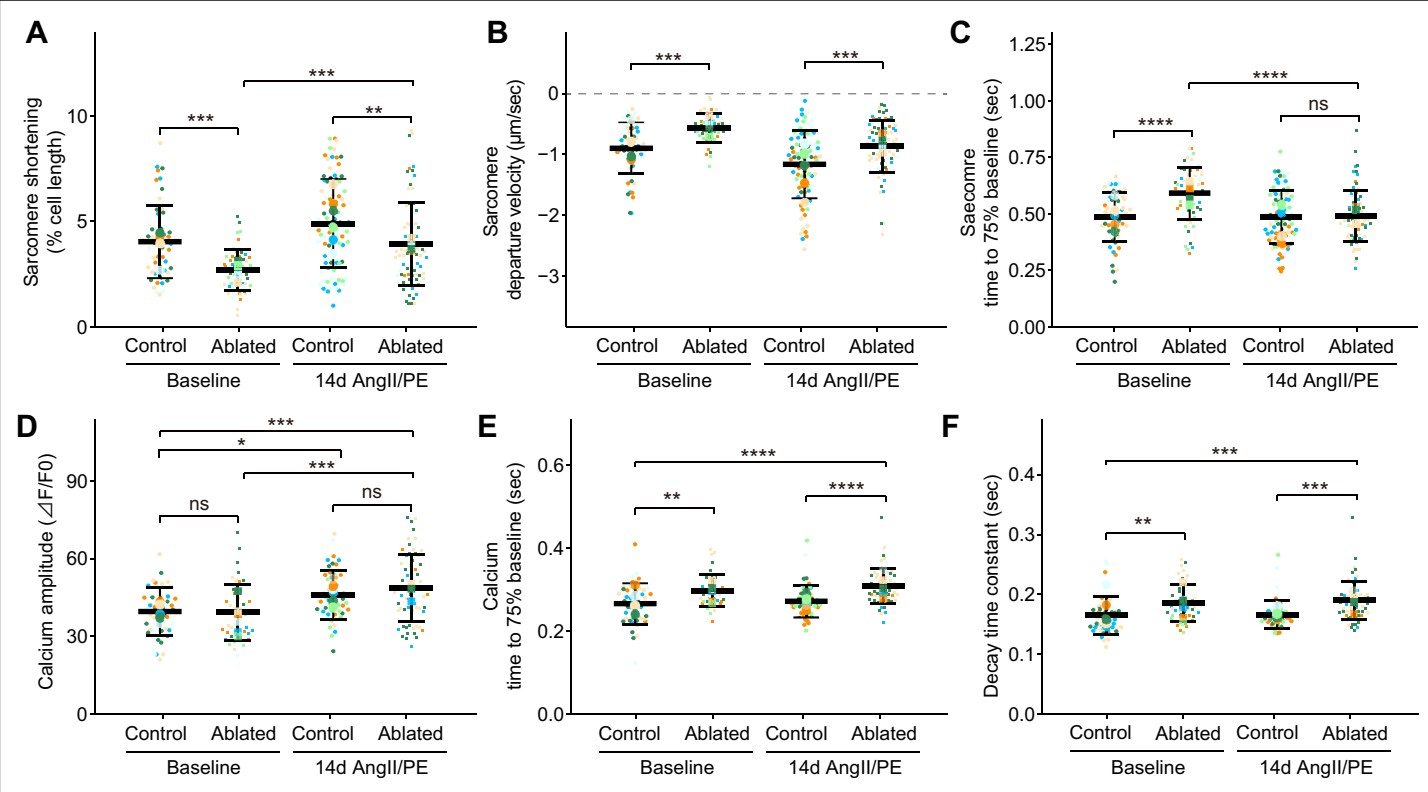

**Figure 6.** Cardiomyocyte contraction and calcium handling after fibroblast ablation. Cardiomyocyte (**A–C**) contraction and (**D–F**) calcium-handling recording by edge detection from control and fibroblast-ablated hearts at baseline and after 14 days of angiotensin II/phenylephrine (AngII/PE) infusion. Small dots represent individual measurements and large dots mean values of each subject colored separately. Results are mean ± SD. Statistical significance was determined by a two-way ANOVA with Tukey's test. Control: n=5, 6; ablated: n=6, 5 (baseline and AngII/PE, respectively). ns: not significant, p>0.05; *p≤0.05; **p≤0.01; ***p≤0.001; ****p≤0.0001.

(*Figure 7C–D*). Interestingly, fibroblast-ablated mice maintained better diastolic function represented by E/E' in the first 2 weeks post-MI but were similar to control mice at 4 weeks (*Figure 7E–F*). Ejection fraction over time demonstrated that fibroblast-ablated mice had retained ejection fraction compared to control mice. These differences were most significant in the first 4 weeks (*Figure 7G*). However, cardiomyocyte CSA (*Figure 7H*) and the area occupied by collagen (*Figure 7I–J*) were similar between control and ablated hearts at 10 weeks post-MI. These results suggest that a reduction in fibroblasts prior to injury does not eliminate replacement fibrosis during repair. Immediately following injury, cardiac output was better, but over time the differences between control and ablated animals were not significant.

To determine if the residual fibroblasts expanded to compensate for fibroblast loss, we examined the number of PDGFRα⁺ and *Col1a1*-expressing fibroblasts in the infarct area at 5 and 10 days post-MI. At 5 days post-MI, fewer PDGFRα⁺ fibroblasts were observed in the lesions of ablated mice; however, the difference between control mice and those with ablated PDGFRα⁺ fibroblasts diminished by day 10 (*Figure 8A–B*). As few ruptures occurred after LAD ligation, we examined infarct areas for collagen-expressing cells, and remarkably, control and ablated lesions exhibited a similar number of *Col1a1*-GFP⁺ cells (*Figure 8A and C*). On day 5, PDGFRα and collagen expression did not overlap extensively in the ablated hearts, but by day 10, the percentage of double-positive PDGFRα⁺/collagen-expressing fibroblasts was similar to controls (*Figure 8A and D*). Within the lesions, total fibroblast content was similar between control and ablated mice (*Figure 8E*). While total fibroblast numbers remained relatively similar, an increase in CD45⁺ cells was detected in the lesions of ablated hearts at 5 days post-MI (*Figure 8F–G*), suggesting a potential inflammatory feedback role for PDGFRα⁺ fibroblasts immediately following MI. Consistent with functional fibroblast and immune cell activation, zymography showed similar proteolytic activity when comparing control and ablated hearts (*Figure 3D*). To obtain a more comprehensive view of the fibroblast numbers after MI, we

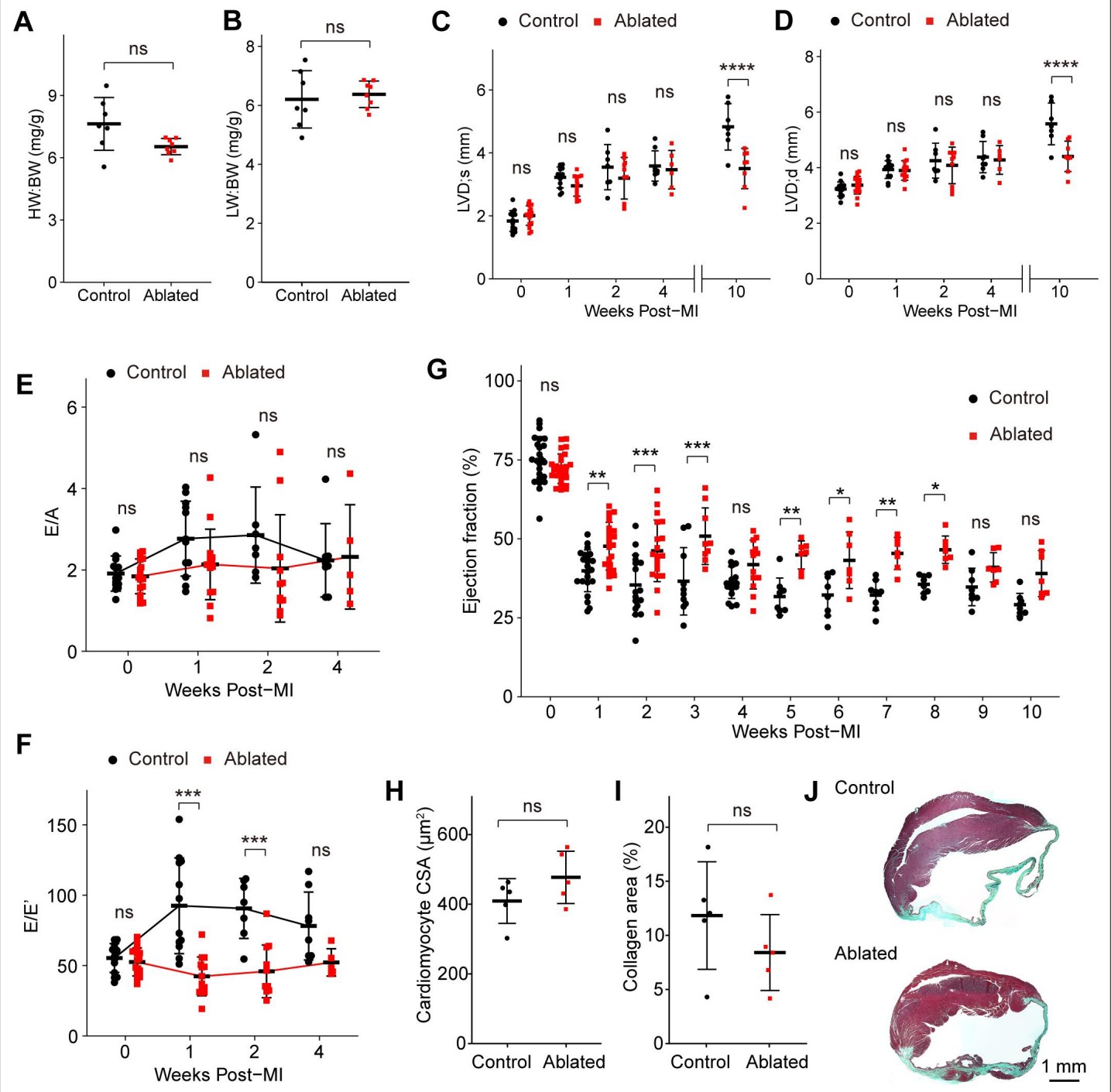

**Figure 7.** Key cardiac parameters and function after myocardial infarction (MI). (**A**) Heart weight to body weight (HW:BW) ratio, (**B**) lung weight to body weight (LW:BW) ratio (control: n=7; ablated: n=8), (**C**) diastolic left ventricle internal diameter (LVID), and (**D**) systolic LVID at each timepoint post-MI. Control: n=14, 12, 7, 8, 7; ablated: n=17, 14, 10, 6, 8 (baseline, 1, 2, 4, and 10 weeks post-MI, respectively). Ratio between mitral E wave and A wave (E/A) and (**F**) E' wave (E/E') representing diastolic function. Control: n=14, 12, 7, 8; ablated: n=17, 14, 10, 6 (baseline, 1, 2, and 4 weeks post-MI, respectively). (**G**) The LV ejection fraction (EF) post-MI. Control: n=24, 22, 17, 10, 16, 8, 8, 8, 8, 8, 8; ablated: n=26, 23, 19, 9, 13, 7, 7, 7, 7, 7, 7 for each timepoint. (**H**) Cardiomyocyte cross-sectional area (CSA) 10 weeks post-MI. Control: n=5; ablated: n=5. (**I**) Quantification of percent collagen stained with Masson trichrome (control: n=5; ablated: n=5), and (**J**) representative images in control and ablated hearts. (**A–J**) >7 weeks post-induction at the time of ligation. Results are mean ± SD. Statistical significance was determined by an unpaired t-test (**A–B, H–I**), or two-way ANOVA (**C–G**). ns: not significant, p>0.05; *p≤0.05; **p≤0.01; ***p≤0.001; ****p≤0.0001.

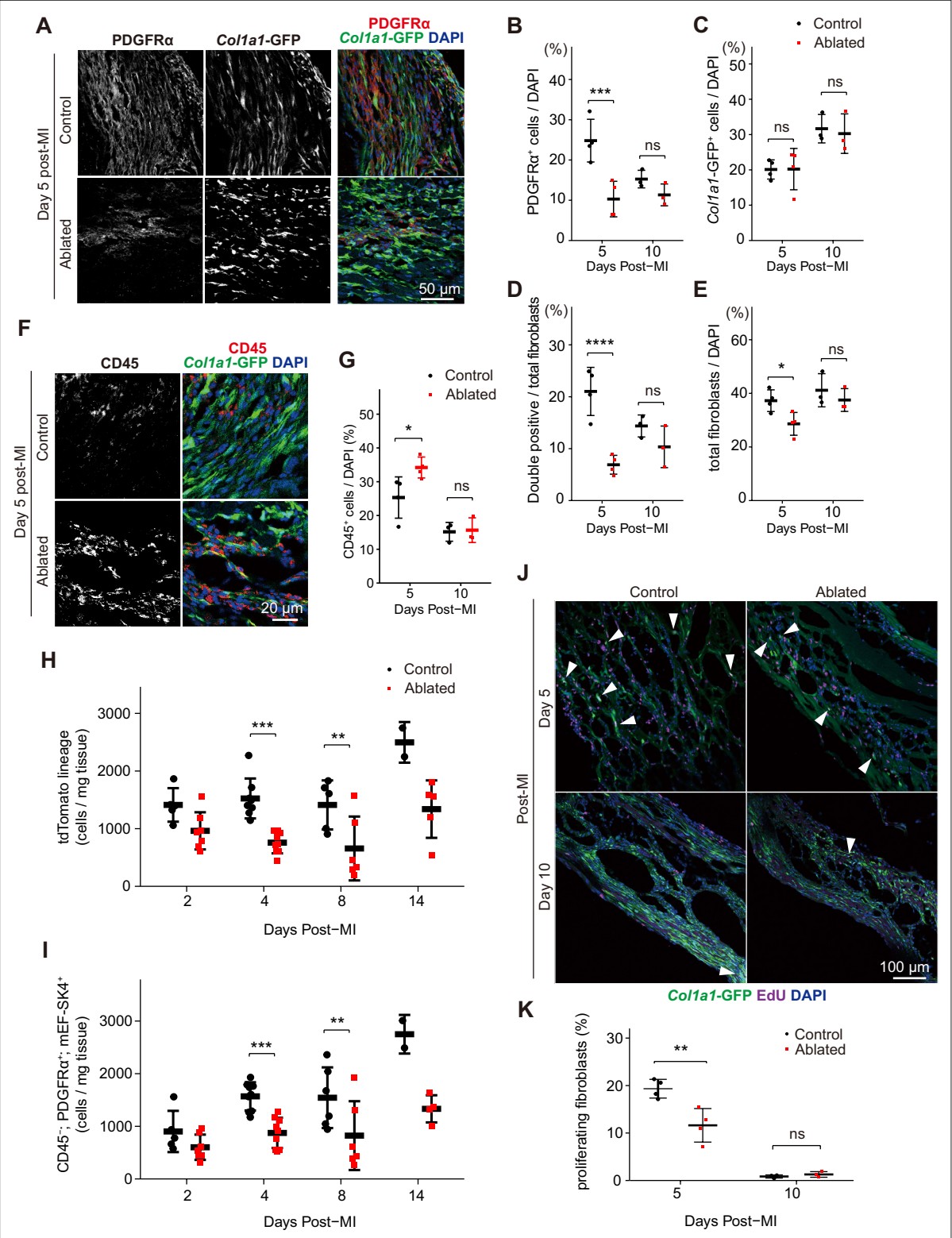

**Figure 8.** Fibroblast characterization after myocardial infarction (MI). (**A**) Representative images of *Col1a1*-GFP and PDGFRα 5 days post-MI. (**B–E**) Quantification of indicated populations normalized to 4′,6-diamidino-2-phenylindole (DAPI)+ cells (**B, C, E**) or total fibroblasts within infarct areas (**D**). Total fibroblasts were calculated as the sum of single positive cells for PDGFRα or *Col1a1*-GFP and double positive cells. Control: n=4, 3; ablated: n=4, 3 (5 and 10 days post-MI, respectively). (**F**) Representative images of *Col1a1*-GFP and CD45 immunostaining at 5 days post-MI. (**G**) Quantification of CD45+ cells. Control: n=4, 3; ablated: n=4, 3 (5 and 10 days post-MI, respectively). (**H–I**) Quantification of (**H**) *Pdgfra*lin (control: n=5, 8, 5, 2; ablated:

*Figure 8 continued on next page*

*Figure 8 continued*

n=7, 9, 6, 5 [2, 4, 8, and 14 days post-MI, respectively]) and (**I**) CD45⁻; PDGFRα⁺; mEF-SK4⁺(control: n=5, 9, 6, 2; ablated: n=7, 9, 6, 4 [2, 4, 8, and 14 days post-MI, respectively]) by flow cytometry at indicated time points post-MI from total ventricle tissue. (**J**) Representative images of *Col1a1*-GFP and EdU at indicated time points post-MI. White arrowheads indicate proliferating GFP⁺ fibroblasts. (**K**) Percentage of proliferating fibroblasts in GFP⁺ fibroblasts. Control: n=4, 3; ablated: n=4, 3 (5 and 10 days post-MI, respectively). Statistical significance was determined by a two-way ANOVA with the Tukey's test (**B–E, G, H, I, K**). ns: not significant, $p > 0.05$; *$p \le 0.05$; **$p \le 0.01$; ***$p \le 0.001$; ****$p \le 0.0001$.

The online version of this article includes the following figure supplement(s) for figure 8:

**Figure supplement 1.** Mural cell contribution after myocardial infarction (MI) (**A, B**) Chondroitin sulfate proteoglycan 4 (CSPG4/NG-2) and (**C**) PDGFRβ lineage expansion at the indicated time points.

performed flow cytometry from whole ventricles and found that PDGFRα lineage and PDGFRα-expressing fibroblast numbers remained lower than those observed in controls (*Figure 8H, I*). These findings suggest that even when PDGFRα fibroblasts are reduced, the residual fibroblasts can respond to injury resulting in fibroblast numbers that are similar to controls within the lesions.

Despite reduced levels of collagen-expressing cells in ablated hearts before the injury, collagen-expressing cells were abundant after the injury. To determine if mural or smooth muscle cells contributed to some of the collagen⁺ populations, we stained for CSPG4/NG-2 and PDGFRβ, conventional mural cell markers. Immunostaining revealed that the proportion of mural cells in the collagen-expressing population was not different between ablated and control hearts post-MI (*Figure 8— figure supplement 1*). To determine if the similar numbers of fibroblasts in control and ablated hearts resulted from increased fibroblast proliferation in the ablated hearts, we compared EdU incorporation (*Figure 8J–K*). Interestingly, among collagen-expressing cells, proliferation was reduced in hearts with reduced fibroblast numbers, suggesting that fibroblast numbers cannot be explained by an increase in the proliferation rates in the remaining fibroblast population. Taken together, these data suggest that the similar number of fibroblasts within the lesion may result by recruitment of existing fibroblasts rather than alternative origins of fibroblasts or increased proliferation.

## Discussion

Our knowledge of fibroblast biology in the injured heart is rapidly expanding. Because fibroblast-targeting is being contemplated as a mitigation strategy in fibrosis-associated cardiovascular diseases, an understanding of the homeostatic roles of fibroblasts is a crucial first step in this process. Fibroblasts are thought to be the primary source of type I collagen, and homeostatic collagen turnover and synthesis in the rat heart is estimated to be between 6 and 9% per day (*Mays et al., 1991*). Yet, our study reveals that the murine adult heart is functionally indistinguishable after sustaining fibroblast reduction for up to 7 months after ablation. Even though a complete ablation of fibroblasts was not obtained, the remaining fibroblasts did not expand and repopulate the heart. Despite this substantial loss of fibroblasts, proteomics of the decellularized ventricles did not detect massive changes in protein abundance, with type I collagen protein remaining relatively normal and fibrillar organization retained. Proteomics analysis was challenged by the high abundance of cardiomyocyte components relative to ECM, ECM post-translational modifications, and limited extractability of cross-linked components. Nonetheless, the collective data suggest that ECM in the unchallenged adult heart, once formed, is stable and has a low turnover rate, consistent with results from multi-isotope imaging mass spectrometry, suggesting that ECM proteins have half-lives on the order of years (*Toyama et al., 2013*).

The significant changes identified by TAILS in fibroblast-ablated hearts were surprisingly few in number and did not include any ECM-derived peptides, despite extensive offline fractionation prior to LC-MS/MS and identification of a large number of internal protein N-termini, suggesting that fibroblast depletion may not dramatically impact ECM proteolysis. One explanation is that although fibroblasts are primarily responsible for ECM degradation and deposition, there may be relatively low turnover at baseline in healthy hearts, and a 60–80% loss of these cells may not result in a significant change in protein composition. Alternatively, it is possible that the high sample complexity introduced by abundant cardiomyocyte proteins may have reduced the yield of ECM peptides, consistent with the finding that of the 1834 peptides, only 4.3% originated in secreted/ECM proteins. Nevertheless, this approach, in addition to label-free quantitation (LFQ)-proteomics and cardiac function analysis,

also supports a relatively mild impact of fibroblast ablation overall on the heart, although significant changes in several contractile proteins and metabolic enzymes were identified. Perhaps, one potent contributor to unaltered ECM turnover is the unexpected reduction of proteolytic activity detected by gelatin zymography. Indeed, in addition to gelatin, MMP2 and MMP9, the enzymes principally detected by gelatin zymography are known to cleave numerous ECM components.

In support of previous findings (*Ivey et al., 2019*), there was no mechanism within the heart that signals fibroblast replenishment even with reductions up to 75%. It has been suggested that cardiac fibroblasts are heterogeneous (*Skelly et al., 2018*; *Farbehi et al., 2019*), and there is a possibility that a PDGFRα-negative fibroblast population compensates for fibroblast loss. However, analysis of total heart RNA argues that type I collagen transcript and reporter transgene levels remain reduced. Thus, in an undamaged heart, a reduction in collagen synthesis might be balanced by an equal reduction in collagen degradation, another action that has been primarily attributed to fibroblasts.

While severe changes to fibrillar collagen were not observed, there were more notable differences in the basement membrane and microfibrillar collagen surrounding cardiomyocytes. Therefore, these data suggest that fibroblasts produce factors, including matricellular proteins that may stabilize the fibrillar collagen network or its interactions with the cardiomyocyte basement membrane. Because collagen VI is secreted by fibroblasts and connects the basement membrane to fibrillar collagen (*Naugle et al., 2006*), we suspected that the observed basement membrane alterations in fibroblast-ablated hearts may be directly due to fibroblast loss. Alternatively, it is also possible that the observed basement membrane changes may be indirectly caused by disruption of collagen matrix organization or that the basement membrane and cell-proximate networks are not as well protected from a turnover as cross-linked collagen. Collagen VI was reported to be overexpressed in hypertension, diabetes, and post-MI (*Naugle et al., 2006*; *Spiro and Crowley, 1993*). Interestingly, a recent study demonstrated that *Col6a1* knockout mice had improved heart function after MI (*Luther et al., 2012*). Therefore, our results are consistent with previous data in that the observed reduction in collagen VI at baseline and after 14 days of AngII/PE could, in part, explain the cardioprotective effect in fibroblast-ablated hearts after injury.

Because contraction is dependent on cardiomyocyte adhesion to the ECM which is mediated by the basement membrane, we expected to observe a disruption of cardiomyocyte contraction in fibroblast-ablated hearts. Indeed, our study is among the first to demonstrate that in vivo modulation of the ECM by fibroblasts leads to altered myocyte function. One explanation is that the disrupted basement membrane in ablated hearts could decrease myofilament calcium sensitivity (*Chung et al., 2016*). While we did not examine free calcium in our hearts, the disrupted basement membrane could alter myofilament adhesion leading to reduced contraction of the sarcomeres. Changes in contraction efficiency could also be a result of the observed altered proteolysis of proteins that are part of the contractile machinery, such as myosin-6. Thin filament proteins can undergo modifications, such as proteolysis leading to thin filament deactivation and slowed myocardial relaxation (*Biesiadecki et al., 2014*). In response to AngII/PE infusion, decreased cardiomyocyte shortening and sustained normal cardiac functions were observed in fibroblast-ablated hearts. Therefore, we hypothesized that fibroblast loss predisposes the heart to physiological, rather than pathological hypertrophy in response to drug-induced fibrosis. However, because we analyzed hearts only after two weeks of AngII/PE infusion, long-term injury models should be examined to determine whether physiological hypertrophy is sustained over time in fibroblast-ablated hearts.

Because activated fibroblasts are key contributors to scar formation, fibroblasts have increasingly become a target of interest in combating heart disease. However, studies that have focused on specifically reducing activated fibroblasts or targeting genes in activated fibroblasts to attenuate pathological fibrosis have produced conflicting results. In some circumstances, fibroblast disruption leads to rupture while in other scenarios manipulations of fibroblasts appear to be protective (*Kanisicak et al., 2016*; *Khalil et al., 2019*; *Molkentin et al., 2017*; *Kaur et al., 2016*; *Aghajanian et al., 2019*; *Travers et al., 2017*; *Kretzschmar et al., 2018*; *Maruyama et al., 2016*; *Kong et al., 2018*). Our work further demonstrates that resident fibroblast reduction prior to injury may have beneficial outcomes under certain pathological conditions. Similarly, depletion of the resident fibroblast population prior to AngII/PE infusion reduced reactive fibrosis; however, replacement fibrosis was not affected after MI. We suspect that the collagen differences between these two injury types may be a result of replacement fibrosis leading to a more rapid and robust fibroblast expansion with MI compared to AngII/

PE. We observed very few ruptures of the ventricular wall, in contrast to what has been previously reported with fibroblast manipulation (*Kanisicak et al., 2016*; *Molkentin et al., 2017*). Several distinctions between our model and those in other reports are that we removed fibroblasts several weeks prior to injury and the remaining fibroblasts are wild-type, while the previous studies were primarily targeting an activated, periostin-expressing fibroblast population. Our results suggest that lowering initial fibroblast levels could have beneficial effects in pathological conditions particularly in response to reactive fibrosis.

While we did not evaluate the ECM protein composition of the fibrotic scar, we predict that the ECM prior to injury may be altered resulting in subtle changes in matrix organization and stiffness. Although our proteomics data did not detect large differences in ECM components of fibroblast-ablated hearts at baseline, the differentially abundant ECM proteins that were observed could contribute to the protective response after injury. Because depletion of fibroblasts occurs several weeks before injury, the heart may have adapted prior to insult. Moreover, fibroblast loss increased cardiomyocyte hypertrophy in response to injury implicating a dynamic interaction between fibroblast and cardiomyocytes during pathology. Further investigation of the cardioprotective effect of resident fibroblast depletion will provide insights into the potential efficacy of anti-fibrotic therapies and delineate the long-term effects on the ECM and cardiac function.

In summary, these studies demonstrate the surprising finding that a significant reduction in cardiac fibroblasts is not detrimental to basal heart function. These observations suggest that cardiac tissue and especially the ECM may be very resilient. Further studies to determine the level of fibroblast loss that can be tolerated by cardiac tissue are warranted. Fibroblast loss prior to injury potentially resulted in a type of pre-conditioning that may be cardioprotective after injury. Further examination of the effect of fibroblast ablation after injury will provide insight into whether manipulation of the fibroblasts at specific stages of the pathological response may also have therapeutic value. Our study reinforces the idea that controlled fibroblast reduction may be a potential strategy for reducing maladaptive fibrosis in heart failure and other sustained cardiac diseases.

## Methods

### Mice

All animal protocols and experiments were approved by the University of Hawaii at Manoa Institutional Animal Care and Use Committee. Both male and female with a mixed C57Bl/6 background were used for these studies. The $Gt(ROSA)26Sor^{tdT}$ (*Madisen et al., 2010*; Jackson labs, 007914) and $Gt(ROSA)26Sor^{DTA}$ (*Wu et al., 2006*; Jackson labs, 010527) mice were purchased from Jackson Laboratory. The *Col1a1-GFP* transgenic mice express cytoplasmic GFP under the control of a *Col1a1* promoter/enhancer and were generated by Dr David Brenner (*Acharya et al., 2012*). The $Pdgfra^{CreERT2/+}$ mice were kindly provided by Dr Brigid Hogan (Duke University Medical Center; *Ivey et al., 2019*). The DTA expression was induced between 8 and 10 weeks of age by oral gavage on two non-consecutive days using 0.2 mg/g of body weight of tamoxifen (AdipoGen, CDX-T0200). The genotype for fibroblast reduction was $Pdgfra^{CreERT2/+}$; $Gt(ROSA)26Sor^{tdT/DTA}$. All control mice used in these experiments were tamoxifen-induced and age-matched. Control genotypes were $Pdgfra^{CreERT2/+}$; $Gt(ROSA)26Sor^{tdT/+}$ or $Pdgfra^{+/+}$; $Gt(ROSA)26Sor^{tdT/DTA}$ (littermate controls). A mix of male and female littermate (Cre⁻) and non-littermate (Cre⁺) controls were used. All experiments were performed on adult mice older than 8 weeks.

### Screening for *Pdgfra* deletion

All mice used in these experiments were screened for fibroblast loss either by PDGFRα protein expression, tdTomato expression, *Col1a1*-GFP transgene expression, or *Pdgfra* transcript levels in the heart or kidney. Kidneys were collected, stored in RNAlater Stabilization Solution (Invitrogen, AM7021), and processed for RNA extraction as described below. Mice with less than 45% reduction of either fibroblasts or *Pdgfra* expression were excluded from studies (*Supplementary file 1e*).

### Immunostaining and microscopy

Cardiac tissue was excised, washed with Dulbeccos phosphate buffered saline (DPBS), and saturated with 3 M KCl to arrest the heart in diastole. Hearts were bisected coronally and fixed with freshly

prepared 4% paraformaldehyde (PFA) for 2 hr at room temperature or 10% neutral buffered formalin (NBF) for 24 hr at 4°C. Tissue fixed with 4% PFA was cryoprotected and frozen embedded. Immunostaining was performed on 10 μm tissue cryosections as previously described (*Ivey et al., 2019*). Primary antibodies used for immunostaining are listed in *Supplementary file 1f*. Primary antibodies were detected using secondary antibodies from Thermo Fisher Scientific at a 1:500 dilution for 1 hr at room temperature. Nuclei were stained with DAPI (Roche, 10-236-276-001). Tissue fixed with 10% NBF was processed for paraffin embedding. Trichrome staining was performed on 5 μm paraffin-embedded tissue sections using Masson or Gomori trichrome stain kit (VWR) according to the manufacturer's protocol. A Zeiss Axiovert 200 microscope equipped with an Olympus DP71 camera was used for imaging. Images and figures were edited in Photoshop CS6 (Adobe) or ImageJ (version 2.3.0/1.53q).

## 5-ethynyl-2'-deoxyuridine (EdU) labeling
Mice were administered with EdU (Lumiprobe, 20,540) by intraperitoneal (IP) injection (50 μg/g) 24 hr before isolation. Hearts were fixed and cryosectioned. The EdU was detected using Click-iT EdU cell proliferation kit (Invitrogen, C10340).

## Western blot
Atria and valves were removed from isolated hearts. The whole ventricle was homogenized in Radio-immunoprecipitation assay buffer with a protease inhibitor cocktail (Bimake, B14001) using a Dounce homogenizer. Samples were centrifuged at 16,000 × *g* for 20 min at 4°C and the supernatant was collected. Blots were probed with primary antibody overnight at 4°C, and then incubated with the corresponding Li-Cor IRDye secondary antibody for 1 hr at room temperature. Primary antibodies used for western blot are listed in *Supplementary file 1f*. An Odyssey CLx imaging system was used for detection and images were analyzed using Image Studio version 5.2.5 software (LI-COR Biosciences).

## Gelatin zymography
Left ventricle tissue was trimmed to 50 mg, followed by homogenization in Triton X-100 lysis buffer (125 mM NaCl, 25 mM Tris-HCl, 1% Triton X-100, pH 8.5) with protease inhibitor cocktail (Bimake, B14001) using a Dounce homogenizer. Samples were centrifuged at 16,000 × *g* for 20 min at 4°C, and the supernatant was collected. About 50 μg of protein were loaded per lane. Novex 10% Zymogram Plus gels (Thermo Fisher, ZY00105BOX) were used, and gelatin zymography was performed according to the manufacturer's instructions.

## Quantitative real-time polymerase chain reaction (qRT-PCR)
RNA isolation was performed on whole ventricles using IBI Isolate DNA/RNA reagent (IBI Scientific, IB47602) and PureLink RNA mini kit (Thermo Fisher, 12183025) according to the manufacturer's instructions. RNA quality and concentration were determined by spectrophotometry using a Nano-Drop 2000 instrument (Thermo Fisher). The cDNA was synthesized using M-MLV Reverse Transcriptase (Sigma, M1302) and random hexamer primer (Thermo Fisher, SO142). The qPCR analysis was performed using PowerUp SYBR Green Master Mix (Thermo Fisher, A25742) and a LightCycler 480 instrument (Roche). Samples were run in triplicate and normalized to *18s* or *Gapdh* expression. The $2^{-\Delta\Delta Ct}$ method was used for determining relative gene expression levels. Primer sequences used for qRT-PCR are listed in *Supplementary file 1g*.

## Flow cytometry
Adult hearts were transcardially perfused with PBS containing 0.9 mM Ca$^{2+}$, and atria and valves were removed. Single-cell suspensions were obtained as previously described (*Pinto et al., 2016*). Briefly, tissue was minced and incubated with collagenase type IV (600 U/mL, Worthington, LS004188) and dispase II (1.2 U/mL, Thermo Fisher, 17105041) in DPBS containing 0.9 mM Ca$^{2+}$ for 45 min at 37°C and filtered through both a 40 μm and then a 30 μm filter to ensure a single-cell suspension. Yellow amine-reactive live/dead cell dye (Thermo Fisher, L34959) or LIVE/DEAD Fixable Violet Dead Cell Stain kit (Thermo Fisher, L34964) was used for live/dead discrimination. Cells were then incubated in Fc block (1:100, Tonbo Bioscience, 70–0161 U500) followed by indicated antibodies and fixed with

1% PFA in flow cytometry buffer. Primary antibodies used for flow cytometry are listed in **Supplementary file 1f**. Accucheck counting beads (Thermo Fisher, PCB-100) were added according to the manufacturer's protocol prior to acquisition. Data were acquired on an LSRFortessa (BD Bioscience) and analyzed using FlowJo software version 10 (BD Bioscience). Cell counts were calculated according to the Accucheck counting bead manufacturer's recommendations and normalized to the weight of individual ventricles.

## Hydroxyproline assay

Hydroxyproline assay (Cell Biolabs, STA-675) was performed on whole ventricles isolated from adult hearts. Samples were homogenized in distilled $H_2O$ and hydrolyzed in 6 N HCl for 20 hr at 95°C. Hydrolyzed contents were processed according to the manufacturer's instructions. The absorbance of the supernatant was read using Molecular Devices SpectraMax M3 microplate reader at 540 nm wavelength.

## SEM on decellularized tissue

The LV tissue was cut into a 3 × 3 × 3 mm cube and fixed in 10% buffered formalin for 24 hr at room temperature. Fixed tissue was washed with distilled $H_2O$ and decellularized in 10% NaOH until the tissue was clear. Decellularized tissue was washed in distilled $H_2O$ twice for 30 min, and then washed overnight at room temperature. Tissue was fixed with 4% tannic acid for 4 hr, post-fixed with 1% osmium tetroxide in 0.1 M sodium cacodylate, dehydrated through an ethanol series, and dried in a Tousimis Samdri-795 critical point dryer. Samples were mounted on aluminum stubs with double-stick carbon tape and coated with gold/palladium in a Hummer 6.2 sputter coater. Samples were viewed and digital images were acquired with a Hitachi S-4800 field emission scanning electron microscope at an accelerating voltage of 5 kV.

## Decellularization of cardiac tissue for mass spectrometry

Whole ventricles from adult mice were cut into 8–10 uniform pieces, washed in distilled $H_2O$ with protease inhibitor for 30 min, and decellularized in 1% sodium dodecyl sulfate in DPBS with protease inhibitor until the tissue was clear. The timing of decellularization was determined by DNA quantification using DAPI staining to confirm the lack of cells in tissue samples. Decellularized cardiac tissue was washed in distilled $H_2O$ with protease inhibitor three times for 5 min at room temperature with light agitation, and then washed overnight to completely remove detergent. Decellularized tissue was placed in 200 μL of Protein Extraction Reagent type 4 with protease inhibitor and sonicated 10 times for 10 s. At least 100 μg of protein was sent to the University of Mississippi Medical Center for mass spectrometry analysis. Mass spectrometry experiments were performed blinded.

## Proteomics of decellularized tissue

Lysates from decellularized heart tissue were obtained as described above and reduced, alkylated, and trypsin-digested into peptides. The peptides were cleaned using a Sep-Pak Vac C18 cartridge (Waters Corporation) and analyzed label-free by LC-MS/MS using a Q Exactive mass spectrometer (Thermo Fisher). A 15 cm Å ~75 μm C18 column (5 μm particles with 100 Å pore size) was used and the nano-ultra-performance liquid chromatography (UPLC )run at 300 nL/min with a 150 min linear acetonitrile gradient (from 5 to 35% B over 150 min; A=0.2% formic acid in water; B=0.2% formic acid in 90% acetonitrile). The MS/MS was set up with the exclusion of 25s, and the samples were run with high-energy collisional dissociation fragmentation at a normalized collision energy of 30% and an isolation width of 2m/z. The resolution setting was 70,000 for target values of the MS at 1e6 ions and in MS2 at a resolution setting of 17,500 for 1e5 ions. Mass spectrometry analyses were performed at the University of Mississippi Medical Center.

The RAW files were analyzed using Proteome Discoverer 2.2 (Thermo Fisher). Precursor mass tolerance was set at 10 ppm and fragment mass tolerance was set at 0.6 Da. Dynamic modification was set to oxidation (+15.995 Da [M]) and static modification was set to carbamidomethyl (+57.021 Da [C]). Samples were searched against the reviewed mouse database downloaded from Uniprot (on November 2018, with 16977 sequences). A strict false discovery rate of 1% was applied. Label-free quantification was done based on precursor ion intensity and normalization was done using the total

peptide amount (from all peptides identified). Proteins were included only if they were identified by at least two high-confidence peptides.

For further visualizations such as multi-scatter plots, PCA, volcano plot, and heat map, normalized abundances from all the samples were imported in Perseus 1.6.5.0. Values were log 2(x) transformed and data were filtered by excluding proteins that were not identified in at least 50% of the samples. The missing values for proteins present in >50%, but not all samples were imputed from the normal distribution feature. Multi-scatter plots and PCA analysis were performed using all proteins, whereas only ECM proteins were used for generating the volcano plot and unsupervised hierarchical clustering. Perseus software was used to perform a t-test with inbuilt multiple hypothesis testing corrections using a permutation-based false discovery rate.

## Heart degradomics using TAILS

A two-step protein extraction method was performed. The first used a commercially available mild detergent T-Per tissue protein extraction reagent (Thermo Fisher) with protease inhibitor cocktail (Roche), and the second used 4 M GuHCl to solubilize the remaining pellet. Samples were prepared for TAILS analysis as previously described (*Martin et al., 2020*). Briefly, heart tissue was homogenized in 0.5 mL of T-Per on ice and centrifuged, the supernatant was collected in a fresh tube (T-Per extract) and 4 M GuHCl containing a protease-inhibitor cocktail was added to the pellet, and incubated further at 4°C for 24 hr on a rotary shaker (GuHCl extract). All subsequent processing was done separately for T-Per and GuHCl extracts. Protein estimation was done using the Bradford assay and 200 µg of protein from each heart was denatured, reduced, and alkylated as per the iTRAQ labeling protocol (SciX). The iTRAQ labels were reconstituted in dimethyl sulfoxide and samples were incubated with iTRAQ labels for 2 hr in the dark. Excess iTRAQ reagent was quenched by incubation with 100 mM ammonium bicarbonate for 30 min in the dark. After labeling, the respective extracts were combined (i.e. T-Per extracts combined with GuHCl extracts) for subsequent handling and MS analysis. After methanol-chloroform precipitation, the pellet was reconstituted in 100 µL of 100 mM NaOH, 50 µL of H2O, and 100 µL of 100 mM HEPES. Samples were digested overnight with trypsin (1:50 ratio) at 37°C. Peptide quantitation was performed using Pierce Quantitative Colorimetric Peptide Assay kit (Thermo Fisher, 23275) and 500 µg of each sample was retained for analysis by MS (pre-TAILS sample; *Figure 3—figure supplement 2A*). The remaining sample was enriched for blocked N-Termini using HPG-ALD polymer to bind reactive (tryptic) N-termini (TAILS). Pre-TAILS and TAILS samples were fractionated by reverse-phase high-performance liquid chromatography. About 32 fractions were collected which were pooled in bins to generate 4 final fractions for analysis on the Orbitrap Fusion Lumos mass spectrometer at the Lerner Research Institute Proteomics and Metabolomics Core. A small portion of the unfractionated sample was also retained and analyzed on the mass spectrometer. C18 clean-up of samples was done prior to LC-MS/MS. Bioinformatics analysis was done and N-termini identified with a false discovery rate of <1% were annotated essentially as recently described (*Martin et al., 2020*).

## MI

Adult mice >22.0 g were subjected to MI as previously described (*Ieda et al., 2009*). Briefly, a thoracotomy was performed between the third and fourth ribs to expose the LAD artery. The proximal LAD artery was permanently ligated using a 7.0 silk suture. Ligation was confirmed by visualization of LV blanching and S-T elevation on the electrocardiogram.

## Osmotic pump implantation

Adult mice >20.0 g were infused with AngII/PE to induce cardiac hypertrophy and fibrosis. Mice were anesthetized with 1–2% isoflurane and mini-osmotic pumps (Alzet, 2001, 2002, or 2004) were implanted subcutaneously. A combination of 1.5 µg/g/day angiotensin II (Calbiochem, 05-23-0101) and 50 µg/g/day phenylephrine hydrochloride (Sigma, P6126) or saline, was infused for 7, 14, or 28 days.

## Echocardiography

Echocardiography was performed using a Vevo 2100 system (VisualSonics) to analyze cardiac function after MI and AngII/PE infusion. Systolic function was analyzed in conscious mice. Briefly, hair removal

cream was applied to the chest to remove all fur. A layer of ultrasound gel was applied to the animal's chest, and the probe was lowered at the parasternal line, 90° between the probe and the heart. B- and M-modes were performed to record 2-dimensional and 1-dimensional transverse cardiac measurements and used to analyze LV function. Independent echocardiography on mice 7 months post-induction at The Ohio State University also found no differences in cardiac measurements between control and fibroblast-ablated hearts.

Diastolic function was measured in mice anesthetized with 4% isoflurane in 100% oxygen. Anesthesia and body temperature were maintained at 1% isoflurane and 37°C, respectively. M-mode in the short axis at the level of papillary muscles was used to measure systolic function. The 4-chamber apical view with color and tissue Doppler mode was used to measure E/A and E/E' ratios. All measurements were performed by an experienced operator blinded to the mouse genotypes.

## Pressure-volume loop

Cardiac hemodynamic measurements were assessed via a closed chest approach using a 1.4 rodent pressure-volume catheter (Transonic) advanced into the left ventricle through the right carotid artery (*Shettigar et al., 2016*). In brief, mice were anesthetized with ketamine (55 mg kg$^{-1}$) plus xylazine (15 mg kg$^{-1}$) in saline solution and placed in the supine position on a heat pad. Following a midline neck incision, the underlying muscles were pulled to expose the carotid artery. Using a 4–0 suture, the artery was tied and the pressure-volume catheter was advanced through the artery into the left ventricle of the heart. After 5–10 min of stabilization, values at baseline and stimulation at varying frequencies (4–10 Hz) were recorded. To measure the beta-adrenergic response, 5 mg kg$^{-1}$ dobutamine was injected IP. All the measurements and analysis were performed on LabChart7 (ADInstruments).

## Cardiomyocyte CSA quantification

Cardiomyocyte boundaries were identified by WGA labeling in 10 µm tissue sections that were processed as described above. Cardiomyocytes were defined by having a circular to oval shape in cross-section, surrounded by circular capillaries. A total of 100 cardiomyocytes in a section were outlined and CSA was calculated using ImageJ (NIH).

## Cardiomyocyte isolation

Cardiomyocytes from whole ventricle tissue were isolated from adult mice using a modified Langendorff-free collagenase digestion protocol (*Ackers-Johnson et al., 2016*). Briefly, mice were anesthetized with an IP injection of tribromoethanol (0.4 mg/g of body weight), the thoracic cavity was opened, and descending aorta was severed. The right ventricle was immediately flushed with EDTA buffer and the ascending aorta was clamped. The heart was excised and the left ventricle was perfused with EDTA and perfusion buffer. The heart was enzymatically digested with 0.5 mg/mL Collagenase Type II (Worthington, LS004176), 0.5 mg/mL Collagenase Type IV (Worthington, LS004188), and 0.05 mg/mL Protease Type XIV (Sigma, P5147). The atria and valves were removed and tissue was teased apart. Cells were dissociated by gentle trituration with a wide-bore pipette tip and cell suspension was filtered through a 100 µM nylon strainer. Cells were allowed to settle by gravity for 20 min, the supernatant was removed, and the cell pellet was resuspended with Tyrode's solution (130 mM NaCl, 5 mM KCl, 10 mM HEPES, 10 mM glucose, 0.5 mM MgCl$_2$, 1.2 mM CaCl$_2$, pH 7.4) for Ca$^{2+}$ transients and contraction measurements.

## Measurement of cardiomyocyte Ca$^{2+}$ transients

Cardiomyocytes were loaded with 2 µM Fura-2 (Thermo Fisher, F1201) in Tyrode's solution for 15 min. Cells were electrically paced at 1 Hz using an IonOptix Myopacer. Measurements were performed using IonWizard version 6.1 software (IonOptix). Images were obtained by a Nikon Eclipse TE2000-U camera. The amplitude of intracellular Ca$^{2+}$ transient was calculated as the difference between peak and diastolic Ca$^{2+}$ levels according to the equation (F–F0)/F0 after subtraction of background fluorescence. The kinetics of Ca$^{2+}$ transient time constants were determined using exponential curve fitting. Measurements were made in more than 45 cardiomyocytes for each group from n=5–6 mice.

## Microarray analysis

Total RNA was isolated as described above. The quality of RNA was assessed by an Agilent 2100 Bioanalyzer and samples with an RNA integrity number value ≥8.0 were used for microarray analysis. The mRNA transcription profile was determined by Clariom S assay for a mouse (Thermo Fisher, 902931). Analysis was performed using Transcriptome Analysis Console software version 4.0.1. The differentially expressed genes with a $\log_2$ (fold change) ≥2 or ≤–2 and p≤0.05 were analyzed by the functional annotation tool in the database for annotation, visualization and integrated discovery (DAVID) version 6.8 program to search for enriched GO terms.

## Metabolic panel

Blood was obtained from the submandibular vein and evaluated using an iStat Chem8+ cartridge (Abbot) using an i-STAT 1 handheld analyzer (Zoetis) according to the manufacturer's recommendations.

## Image quantification

Band intensity of western blots was measured using ImageJ (version 2.3.0/1.53q). Subtracted background intensity was determined for the same gel image. Fluorescent images were processed and measured using ImageJ. Split channel images were generated, followed by binarization. Fused cells were divided by the watershed function. The number of cells was counted with the analyze particle function or Cell Counter plugin. The identical setting of thresholds was used for each session through automated macros.

## Statistical analysis

All statistical analyses were conducted using Prism 8 (Graphpad Software). A Shapiro-Wilk test was performed to determine the data distribution. If the distribution was normal, an unpaired t-test was used for two group comparison (parametric), otherwise, a Mann-Whitney U test was used as a non-parametric analysis. For multiple comparisons with normal distribution, a one- or two-way ANOVA with the Tukey's test were performed. Statistical variability is expressed as mean ± SD; ns: not significant, p>0.05; *p≤0.05; **p≤0.01; ***p≤0.001; ****p≤0.0001. N-values for experiments are indicated in the figure or figure legends.

## Study approval

All mouse experiments were performed according to the animal experimental guidelines issued and approved by the Institutional Animal Care and Use Committees of the University of Hawaii at Manoa (APN12-1421 and APN12-1469) and The Ohio State University Wexner Medical Center (#2021 A00000070).

## Acknowledgements

This work was supported by NIH HL074257 (MDT), NHLBI Institutional Cardiology Training Grant T32 HL115505 (JTK), and American Heart Association Grants PRE29630019 (JTK) and GRNT33660474 (MDT), by the Allen Distinguished Investigator Program, through support made by The Paul G Allen Frontiers Group and the American Heart Association (SSA), and Japan Society for the Promotion of Science Overseas Research Fellowships (AH). The Fusion Lumos Instrument at the Lerner Research Institute was purchased via an NIH shared instrument grant, 1S10OD023436-0. We thank S Sebastian and C Applegarth for their excellent technical support. We also thank the JABSOM Histology and Imaging Core supported by RCMI-BRIDGES G12 MD007601; the Center for Cardiovascular Research Animal Physiology Core supported by NIH grant P30 GM103341; the UH Cancer Center Genomics and Bioinformatics Shared Resource supported by the NCI Cancer Center Support Grant (CCSG) P30 CA071789; The Microscopy, Imaging, and Flow Cytometry Core at UHCC was supported by SIG: NIH S10OD028515; and the Pacific Biosciences Research Center's Biological Electron Microscope Facility at the University of Hawaii at Manoa.

# Additional information

## Funding

| Funder | Grant reference number | Author |
|---|---|---|
| National Institutes of Health | HL074257 | Michelle D Tallquist |
| National Institutes of Health | HL115505 | Jill T Kuwabara |
| American Heart Association | PRE29630019 | Jill T Kuwabara |
| American Heart Association | GRNT33660474 | Michelle D Tallquist |
| American Heart Association | PRE834732 | Jasmine Chen |
| American Heart Association | | Suneel S Apte |
| Paul G. Allen Frontiers Group | | Suneel S Apte |
| Japan Society for the Promotion of Science | | Akitoshi Hara |
| Naito Foundation | | Akitoshi Hara |

The funders had no role in study design, data collection and interpretation, or the decision to submit the work for publication.

## Author contributions

Jill T Kuwabara, Lydia P DeAngelo, Investigation, Visualization, Methodology, Writing - original draft, Writing - review and editing; Akitoshi Hara, Formal analysis, Validation, Investigation, Visualization, Methodology, Writing - original draft, Writing - review and editing; Sumit Bhutada, Formal analysis, Investigation, Visualization, Methodology, Writing - original draft, Writing - review and editing; Greg S Gojanovich, Investigation, Methodology, Writing - original draft; Jasmine Chen, Kanani Hokutan, Vikram Shettigar, Investigation, Visualization, Methodology, Writing - review and editing; Anson Y Lee, Investigation, Writing - review and editing; Jack R Heckl, Investigation, Visualization, Methodology, Writing - original draft; Julia R Jahansooz, Dillon K Tacdol, Mark T Ziolo, Investigation, Methodology, Writing - original draft, Writing - review and editing; Suneel S Apte, Conceptualization, Supervision, Funding acquisition, Investigation, Methodology, Writing - original draft, Writing - review and editing; Michelle D Tallquist, Conceptualization, Resources, Supervision, Funding acquisition, Validation, Investigation, Visualization, Methodology, Writing - original draft, Project administration, Writing - review and editing

## Author ORCIDs

Sumit Bhutada http://orcid.org/0000-0002-5274-5122
Lydia P DeAngelo http://orcid.org/0000-0002-0549-325X
Suneel S Apte http://orcid.org/0000-0001-8441-1226
Michelle D Tallquist http://orcid.org/0000-0002-1383-144X

## Ethics

All mouse experiments were performed according to the animal experimental guidelines issued and approved by Institutional Animal Care and Use Committees of the University of Hawaii at Manoa (APN12-1421 and APN12-1469) and The Ohio State University Wexner Medical Center (#2021A00000070). All surgeries were performed under isofluorane anesthesia, and every effort was made to minimize suffering.

## Decision letter and Author response

Decision letter https://doi.org/10.7554/eLife.69854.sa1
Author response https://doi.org/10.7554/eLife.69854.sa2

## Additional files

### Supplementary files

• Supplementary file 1. Supplementary tables. (a) PDGFRβ⁺ population in *Col1a1*-GFP+ cells counted by flow cytometry. (b) Metabolic blood panel. (c) Potential proteolytically cleaved (internal) peptides with statistically significant higher abundance in control hearts. (d) Potential proteolytically cleaved peptides with statistically significant higher abundance in fibroblast-ablated hearts. *Eghbali et al., 1988* (e) Efficiency of *Pdgfra* deletion determined for each experiment (expressed as the percent of cells deleted). (f) Cell-specific reagents for tissue/cell staining. (g) Primers used for qRT-PCR.

• MDAR checklist

### Data availability

Mass spectrometry proteomics data have been deposited to the ProteomeXchange Consortium via the PRIDE partner repository with the dataset identifier PXD021741 (shotgun proteomics) and PXD021739 (N-terminomics). All data and materials generated or analyzed during this study are included in this manuscript or deposited in the above repository.

The following previously published datasets were used:

| Author(s) | Year | Dataset title | Dataset URL | Database and Identifier |
|---|---|---|---|---|
| Kuwabara JT, Bhutada S, Shettigar V, Gojanovich GS, DeAngelo LP, Heckl JR, Jahansooz JR, Tacdol DK, Ziolo MT, Apte SS, Tallquist MD | 2020 | Consequences of fibroblast ablation in adult murine hearts | https://doi.org/10.6019/PXD021741 | ProteomeXchange Consortium, 10.6019/PXD021741 |
| Bhutada S, Shettigar V, Gojanovich GS, DeAngelo LP, Heckl JR, Jahansooz JR, Tacdol DK, Ziolo MT, Apte SS, Tallquist MD, Kuwabara JT | 2020 | Consequences of fibroblast ablation in adult murine hearts | https://doi.org/10.6019/PXD021739 | ProteomeXchange Consortium, 10.6019/PXD021739 |

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
