## [Editor Report]

This study is directed at cardiac fibrotic change, which is widespread both under conditions of both normal development and pathological responses, and it is associated with both pump failure and arrhythmic change. A new genetic platform is introduced, in which such processes are reduced. What emerges is a reduction in ventricular, atrial and septal fibroblast density. This study is extremely well done, rigorous, and offers valuable insight for investigators interested in fibrosis, cardiac fibroblast biology, and mechanisms of extracellular matrix remodeling.

---

## [Decision Letter]

**Decision letter after peer review:**

Thank you for submitting your article "Consequences of PDGFRalpha+ fibroblast reduction in adult murine hearts" for consideration by *eLife*. Your article has been reviewed by 3 peer reviewers, including Christopher L-H Huang as Reviewing Editor and Reviewer #1. and the evaluation has been overseen by a Reviewing Editor and a Senior Editor.

The reviewers have discussed their reviews with one another, and the Reviewing Editor has drafted this letter to help you prepare a revised submission.

Essential revisions:

The reviewers are positive about the work, but between them, generate a large number of significant revisions, likely because of the unexpected findings in this study. This is reflected in one of the evaluation studies which suggest that "This study is extremely well done, rigorous, and offers valuable insight for investigators interested in fibrosis, cardiac fibroblast biology, and mechanisms of extracellular matrix remodeling", but however, "….the finding is completely unexpected and the paper…. does not extend to the mechanistic depth is needed to understand the basis for the finding. For this reason this review includes all the detailed comments made by the reviewers, in addition to drawing attention to each of the major points indicated particularly from reviewers 2-3.

Reviewer 2:

1) In some cases, the results are expressed a "stained area/tissue area", when clearly it would have been more appropriate to provide cell counts. At least, they should provide FACS plots showing the gating used for quantification to reassure the readers that PDGFRa was expressed at comparable levels in mice of different ages. Given that it is now widely accepted that cardiac fibroblasts are heterogenous in many ways, it would be important to understand whether this ablation affects this heterogeneity.

2) The hearts contained somewhere between half and one third of their normal number of fibroblasts, suggesting that one or two cell divisions would be sufficient to bring these cells back to normal levels. Following an MI, these cells enter rapid proliferation, and their numbers increase significantly. What happens in the ablated heart to fibroblast numbers?

3) A time resolved analysis of PDGFRa cells in the context of the two damage models would be very informative and possibly hint at mechanism.

4) Does the increase in NG2 indicate an upregulation of the protein in mural cells or increase (maybe compensatory) in the numbers of mural cells? Adding mural cells to the FACS-based quantification of inflammatory, endothelial and fibroblastic cells would help in answering this.

5) The authors did not attempt to measure diastolic function in the fibroblast-ablated hearts following damage in either the MI or the Ang II/phenylephrine models. The parameters they report are more relevant to systolic function (ejection fraction) and chamber dimension (LVID). However, I would have expected some measurement of transmitral flow such as E/A ratio or E wave deceleration time from a 4- chamber apical view, or even IVRT. This is important as ECM/fibrotic changes are expected to affect LV compliance/stiffness (i.e. diastolic function) usually before systolic involvement.

6) How was the collagen area measured in the 2 damaged models? By antibody staining or using Trichrome staining?

7) An important pathway that regulates calcium sensitivity and the actin cytoskeleton at the same time is the RhoA/Rho kinase pathway. Did the authors find any changes in expression of the proteins in this pathway?

Reviewer 3:

8) It is fascinating that no rupture is observed and that the relative collagen fraction between the ablated mice and controls is largely similar by 10 days. (A) Have the non-ablated fibroblasts proliferated and normalized the fibroblast numbers back to wild type levels? (B) Is it possible that collagen area as a percentage of tissue area may not be the most informative metric of fibrosis due to the significantly altered cardiac dimensions in mice with ablated fibroblasts? (C) It is puzzling that fibroblast ablation has so little impact on the ECM opening up questions regarding the cell state heterogeneity of the non ablated fibroblasts populations- are they all behaving like activated fibroblasts or matrifibrocytes to maintain the matrix?

9) This study paid a lot of attention to mechanisms of ECM deposition and remodeling but very little attention to degradation and turnover of the matrix. Are these mechanisms inhibited such that the matrix is no longer turning over once fibroblasts are ablated?

10) What is the basis for the reduction in collagen deposition observed for the Ang/PE model but not for the infarction model?

11) If the TAILS proteomics data suggests that cardiomyocyte remodeling may be occurring, is this finding reflected in the geometry of isolated myocytes at baseline measured for their contraction characteristics in figure 7?

*Reviewer #1 (Recommendations for the authors):*

(a) In the summary, there are a few ambiguities that slightly obscure the meaning: the term 'uninjured' is unclear; the following sentence ["Analysis of cardiomyocyte function demonstrated weaker 42 contractions and slowed calcium decline in both uninjured and AngII/PE infused 43 fibroblast-ablated mice."] apparently differs from the comment that ", cardiac function was better 40 preserved following angiotensin II/phenylephrine (AngII/PE)-induced fibrosis and 41 myocardial infarction".

(b) The Introduction is an appropriate review of the role of fibroblasts in cellular matrix formation and matrix formation, but should be slightly revised to provide a brief paragraph on the relationships between fibroblasts and the cardiomyocytes themselves, particularly in relation to influencing gap junction cohesion as well as, through their fusion, effects on capacitative properties of the cardiomyocyte syncytium with a potential for arrhythmic substrate.

(c) When the discussing the Results, the following merit further discussion.

(i) This appeared to be a system that did not show pathological hypertrophy: it exhibited downregulated cell adhesion, collagen binding, collagen fibril organization, and ECM organization, and upregulated membrane, mitochondrial and Z-disc genes, but an absence of uupregulation in Nppa, Nppb, and Myh7 and Myh6 genes related to pathological hypertrophy.

(ii) There was a contrast between: Normal indicators of excitation contraction coupling with similar ca^2+^ transients in control and ablated cardiomyocytes both before and following AngII/PE infusion, with greater transients in the latter, in contrast to:

(iii) Mechanical indicators: Cardiomyocytes from ablated hearts showing reduced baseline sarcomeric shortening, speed of contraction with and slowed relaxation.

*Reviewer #2 (Recommendations for the authors):*

The authors use an induction protocol that seems to lead to an efficiency of CRE activation somewhat below what has been reported for similar strains by others. They use a threshold of at least 45% reduction to include individual animals in the analysis. This does not seem a major improvement over their recent work in which they report the effects of a 50% reduction is minimal. Also, if their threshold is 45%, why do they keep referring to a 60 to 80% ablation in the text?

One important and surprising finding in this paper is that the cells surviving the ablation did not expand to make up for the lost numbers. However, I was surprised to see that most of the evaluation of the extent of depletion over time relied on the expression of PDGFRa itself, a gene whose expression has been reported to be modulated, that is expressed at different levels in different subsets (based on published scRNAseq data) and that is expressed by a single allele in this experimental setup. In some cases, the results are expressed a "stained area/tissue area", when clearly it would have been more appropriate to provide cell counts. At least, they should provide FACS plots showing the gating used for quantification to reassure the readers that PDGFRa was expressed at comparable levels in mice of different ages. Given that it is now widely accepted that cardiac fibroblasts are heterogenous in many ways, it would be important to understand whether this ablation affects this heterogeneity. In other words, does ablation preferentially remove subpopulations of fibroblasts (such as more proliferative cells that express higher levels of Pdgfra) more than others? Knowledge of this may explain possible reasons for the lack of replenishment of fibroblasts. I think it would be worth using scRNAseq in this context.

The CRE strain used is active in a number of other organs and anatomical locations, ranging from lung fibroblast to adipogenic cells in fat depots to oligodendrocyte progenitors in the CNS. Did the authors look into these organs, or notice any systemic effects of their depletion protocol?

Based on the data presented, the hearts contained somewhere between half and one third of their normal number of fibroblasts, suggesting that one or two cell divisions would be sufficient to bring these cells back to normal levels. Following an MI, these cells enter rapid proliferation, and their numbers increase significantly. What happens in the ablated heart to fibroblast numbers? Do they increase as expected, do they plateau at lower or similar levels to the ones observed in controls? Does the scar still contain the same number of collagen-expressing cells?

A time resolved analysis of PDGFRa cells in the context of the two damage models would be very informative and possibly hint at mechanism, and I was surprised to see the analysis limited to one time point and performed by quantifying genes rather than cells themselves.

The proteomic analysis of the de-cellularized heart is a strong point of the paper. However, some of the changes observed may not be a direct effect of fibroblast ablation. For example, one of the proteins the authors reported to be upregulated in the fibroblast-ablated hearts is NG2, which is not expressed to any significant extent in fibroblasts. Rather, it is expressed at high levels in mural cells such as pericytes and vascular smooth muscle cells. Does the increase in NG2 indicate an upregulation of the protein in mural cells or increase (maybe compensatory) in the numbers of mural cells? Adding mural cells to the FACS-based quantification of inflammatory, endothelial and fibroblastic cells would help in answering this.

I am a bit confused by the fact that the authors, although reported decreased LV internal diameter, found that cardiomyocytes did not increase in size. Usually, the reason for decreased chamber size would be hypertrophy, especially if it is observed in systole and diastole. How do the authors explain this?

I am surprised that the authors did not attempt to measure diastolic function in the fibroblast-ablated hearts following damage in either the MI or the Ang II/phenylephrine models. The parameters they report are more relevant to systolic function (ejection fraction) and chamber dimension (LVID). However, I would have expected some measurement of transmitral flow such as E/A ratio or E wave deceleration time from a 4- chamber apical view, or even IVRT. This is important as ECM/fibrotic changes are expected to affect LV compliance/stiffness (i.e. diastolic function) usually before systolic involvement.

How was the collagen area measured in the 2 damaged models? By antibody staining or using Trichrome staining? Why in the authors' opinion did fibroblast ablation decrease collagen area in the Ang II/phenylephrine model but not the MI model? Again these questions may be answered by analyzing the proliferative response to these insults.

A decrease in shortening without a decrease in calcium transient amplitude, as seen in the ablated hearts, indicates calcium desensitization. An important pathway that regulates calcium sensitivity and the actin cytoskeleton at the same time is the RhoA/Rho kinase pathway. Did the authors find any changes in expression of the proteins in this pathway?

*Reviewer #3 (Recommendations for the authors):*

In this manuscript, Kuwabara and colleagues use genetic ablation to reduce the number of fibroblasts resident to the heart. At baseline, the authors observe that fibroblast numbers stay proportionally low after ablation, but with very minimal effects to the structure or composition of the extracellular matrix. Fibroblast ablation prior to myocardial infarction is shown to be beneficial to cardiac function without affecting relative abundance of scar tissue, whereas in an Ang/PE model of fibrosis collagen deposition is impaired and systolic function is preserved.

Major Concerns

1. For the myocardial infarction model presented in Figure 4, it is fascinating that no rupture is observed and that the relative collagen fraction between the ablated mice and controls is largely similar by 10 days. (A) Have the non-ablated fibroblasts proliferated and normalized the fibroblast numbers back to wildtype levels? (B) Is it possible that collagen area as a percentage of tissue area may not be the most informative metric of fibrosis due to the significantly altered cardiac dimensions in mice with ablated fibroblasts? (C) It is puzzling that fibroblast ablation has so little impact on the ECM opening up questions regarding the cell state heterogeneity of the non ablated fibroblasts populations- are they all behaving like activated fibroblasts or matrifibrocytes to maintain the matrix?

2. This study paid a lot of attention to mechanisms of ECM deposition and remodeling but very little attention to degradation and turnover of the matrix. Are these mechanisms inhibited such that the matrix is no longer turning over once fibroblasts are ablated?

3. What is the basis for the reduction in collagen deposition observed for the Ang/PE model but not for the infarction model?

4. If the TAILS proteomics data suggests that cardiomyocyte remodeling may be occurring, is this finding reflected in the geometry of isolated myocytes at baseline measured for their contraction characteristics in figure 7?

---

## [Author Response]

Essential revisions:The reviewers are positive about the work, but between them, generate a large number of significant revisions, likely because of the unexpected findings in this study. This is reflected in one of the evaluation studies which suggest that "This study is extremely well done, rigorous, and offers valuable insight for investigators interested in fibrosis, cardiac fibroblast biology, and mechanisms of extracellular matrix remodeling", but however, "….the finding is completely unexpected and the paper…. does not extend to the mechanistic depth is needed to understand the basis for the finding. For this reason this review includes all the detailed comments made by the reviewers, in addition to drawing attention to each of the major points indicated particularly from reviewers 2-3.

Regarding a better understanding of the fibroblast responses after injury we have extensively updated the myocardial section including data in the first two weeks after injury. These data include proliferation results, investigation of PDGFRb expressing cells, measurement of diastolic function, quantification of cells by flow, and examination of collagen expressing cells in infarcts. The specific revisions related to reviewers’ queries are described below:

Reviewer 2:1) In some cases, the results are expressed a "stained area/tissue area", when clearly it would have been more appropriate to provide cell counts. At least, they should provide FACS plots showing the gating used for quantification to reassure the readers that PDGFRa was expressed at comparable levels in mice of different ages. Given that it is now widely accepted that cardiac fibroblasts are heterogenous in many ways, it would be important to understand whether this ablation affects this heterogeneity.

We have now included the gating strategy for PDGFRa expressing cells in Figure 1 supplement 1. The flow cytometry demonstrates a relatively constant level of PDGFRa expression, independent of age, and the technique is sensitive to surface expression as we observed a decreased mean fluorescence intensity when animals were heterozygous for PDGFRa (as in the PDGFRa CreERT2/+ mice). As the reviewer was concerned about fibroblast heterogeneity, for updated analyses we have used Col1a1-GFP as an unbiased tag of collagen producing cells. The collagen GFP quantification identifies cells with type I collagen production and provides a non-PDGFRa focused view of the hearts at baseline, with aging, and after injury Figures 1A-E.

In regard to cell counts, in new data we have included cell counts/nuclei (Figure 1E, Figure 8 B, C, E). We agree with the reviewer that it is difficult to normalize cell counts. After injury, when quantifying fibroblasts, distinguishing individual fibroblasts, which have significant cellular extensions and increase in volume and can amass in one location is difficult. In addition, immune cell infiltration can vary greatly, altering the relative cell counts. In areas with cardiomyocytes, nuclear numbers can be skewed due to cardiomyocyte area. As these technical issues are most prevalent during MI and can confound quantification by IHC, we have included flow cytometry data to provide an additional perspective on cell number before and after MI Figures 1F and 8H,I.

2) The hearts contained somewhere between half and one third of their normal number of fibroblasts, suggesting that one or two cell divisions would be sufficient to bring these cells back to normal levels. Following an MI, these cells enter rapid proliferation, and their numbers increase significantly. What happens in the ablated heart to fibroblast numbers?

We have now included information documenting fibroblast numbers from multiple perspectives with several time points after LAD ligation. The new data is included in new figure 8. These data demonstrate that the remaining fibroblasts expand, but do not reach the levels observed in the control hearts.

3) A time resolved analysis of PDGFRa cells in the context of the two damage models would be very informative and possibly hint at mechanism.

We have now provided additional time points addressing cell numbers and fibroblast proliferation in the first two weeks after injury including both PDGFRa and collagen expressing cells after injury in Figure 8 and Figure 4 supplemental figure 1.

4) Does the increase in NG2 indicate an upregulation of the protein in mural cells or increase (maybe compensatory) in the numbers of mural cells? Adding mural cells to the FACS-based quantification of inflammatory, endothelial and fibroblastic cells would help in answering this.

To address this reviewer’s query regarding mural cells, we have included stains for NG-2 and/or PDGFRb at baseline and after MI to investigate mural cell expansion. There appears to be limited mural cell expansion in the collagen type I producing population. Table 1, Figure 1 supplemental figure 3 and Figure 8 supplemental figure 1.

5) The authors did not attempt to measure diastolic function in the fibroblast-ablated hearts following damage in either the MI or the Ang II/phenylephrine models. The parameters they report are more relevant to systolic function (ejection fraction) and chamber dimension (LVID). However, I would have expected some measurement of transmitral flow such as E/A ratio or E wave deceleration time from a 4- chamber apical view, or even IVRT. This is important as ECM/fibrotic changes are expected to affect LV compliance/stiffness (i.e. diastolic function) usually before systolic involvement.

We have now performed E/E’ and E/A measurements after permanent ligation of LAD as well as in old control and ablated mice. We observe a significant difference between control and ablated hearts after MI suggesting that reducing the number of fibroblasts helps retain heart diastolic function, but even after a year of deletion ablated mice have similar heart function to controls in the absence of injury Figure 7 and Figure 1 supplement figure 2.

6) How was the collagen area measured in the 2 damaged models? By antibody staining or using Trichrome staining?

We have now indicated that collagen area was measured using Masson’s or Gomori trichrome stain in the Materials and methods.

7) An important pathway that regulates calcium sensitivity and the actin cytoskeleton at the same time is the RhoA/Rho kinase pathway. Did the authors find any changes in expression of the proteins in this pathway?

Our microarray analysis documented no change in expression of RhoA/Rho kinase either at baseline or after angiotensin/PE. We performed Reactome pathway analysis with the microarray data of baseline and AngII cohorts and observed no significant terms associated with Rho, RhoA, or Rho-GTPase.

Reviewer 3.8) It is fascinating that no rupture is observed and that the relative collagen fraction between the ablated mice and controls is largely similar by 10 days. (A) Have the non-ablated fibroblasts proliferated and normalized the fibroblast numbers back to wild type levels?

We performed EdU labeling at day 5 and day 10 after MI and surprisingly observed lower rates of proliferation, in line with an overall reduced number of fibroblasts after MI Figure 8 J,K.

(B) Is it possible that collagen area as a percentage of tissue area may not be the most informative metric of fibrosis due to the significantly altered cardiac dimensions in mice with ablated fibroblasts?

This is entirely possible, but for the purpose of this manuscript we simply wanted to demonstrate that remaining fibroblasts can compensate in an injury scenario.

(C) It is puzzling that fibroblast ablation has so little impact on the ECM opening up questions regarding the cell state heterogeneity of the non ablated fibroblasts populations- are they all behaving like activated fibroblasts or matrifibrocytes to maintain the matrix?

To address this question, we performed qPCR on the adherent cells from ablated hearts and compared their gene expression to controls and at baseline and we observed no obvious compensatory gene expression from the adherent cells. Postn, Chad, and Cthrp1 were not detected in these cultures. The data even demonstrates a decrease in many fibroblast genes, suggesting that the adherent cells may not have a definitive fibroblast gene profile. Possibly smooth muscle cells and pericytes could bias the gene expression in these cultures Figure 1J.

9) This study paid a lot of attention to mechanisms of ECM deposition and remodeling but very little attention to degradation and turnover of the matrix. Are these mechanisms inhibited such that the matrix is no longer turning over once fibroblasts are ablated?

To address this question we included zymography from total heart and observed a consistent reduction in *MMP2* and MMP9 presence supporting the notion that matrix is no longer turned over Figure 3D.

10) What is the basis for the reduction in collagen deposition observed for the Ang/PE model but not for the infarction model?

We suspect the difference is because of the type of insult. With MI there is a rapid inflammatory response with an exponential burst of fibroblasts in a matter of days, while the angII/PE is more gradual with less fibroblast expansion, relatively. We add this speculation to the discussion page 24.

11) If the TAILS proteomics data suggests that cardiomyocyte remodeling may be occurring, is this finding reflected in the geometry of isolated myocytes at baseline measured for their contraction characteristics in figure 7?

We did not note any overt shape changes in the isolated cardiomyocytes.

Reviewer #1 (Recommendations for the authors):(a) In the summary, there are a few ambiguities that slightly obscure the meaning: the term 'uninjured' is unclear; the following sentence ["Analysis of cardiomyocyte function demonstrated weaker 42 contractions and slowed calcium decline in both uninjured and AngII/PE infused 43 fibroblast-ablated mice."] apparently differs from the comment that ", cardiac function was better 40 preserved following angiotensin II/phenylephrine (AngII/PE)-induced fibrosis and 41 myocardial infarction".

The term uninjured was removed from the text of the manuscript. Also, to address reviewer’s two concerns regarding terminology, we have now changed our statements to reflect the cardiomyocyte measurement being performed. Thus, we have removed terms such as weaker contractions.

(b) The Introduction is an appropriate review of the role of fibroblasts in cellular matrix formation and matrix formation, but should be slightly revised to provide a brief paragraph on the relationships between fibroblasts and the cardiomyocytes themselves, particularly in relation to influencing gap junction cohesion as well as, through their fusion, effects on capacitative properties of the cardiomyocyte syncytium with a potential for arrhythmic substrate.

We have included a brief statement regarding known fibroblast-cardiomyocyte interactions in the introduction.

(c) When the discussing the Results, the following merit further discussion.(i) This appeared to be a system that did not show pathological hypertrophy: it exhibited downregulated cell adhesion, collagen binding, collagen fibril organization, and ECM organization, and upregulated membrane, mitochondrial and Z-disc genes, but an absence of uupregulation in Nppa, Nppb, and Myh7 and Myh6 genes related to pathological hypertrophy.(ii) There was a contrast between: Normal indicators of excitation contraction coupling with similar ca^2+^ transients in control and ablated cardiomyocytes both before and following AngII/PE infusion, with greater transients in the latter, in contrast to:(iii) Mechanical indicators: Cardiomyocytes from ablated hearts showing reduced baseline sarcomeric shortening, speed of contraction with and slowed relaxation.

We observed that the ablated and control mice had increased LV chamber sizes in response to MI. However, ablated mice resisted further dilation of the chamber and remained constant till the end of the study. We have made changes in the text to avoid this confusion. Other queries were addressed above.

Reviewer #2 (Recommendations for the authors):The authors use an induction protocol that seems to lead to an efficiency of CRE activation somewhat below what has been reported for similar strains by others. They use a threshold of at least 45% reduction to include individual animals in the analysis. This does not seem a major improvement over their recent work in which they report the effects of a 50% reduction is minimal. Also, if their threshold is 45%, why do they keep referring to a 60 to 80% ablation in the text?

To our knowledge, no previous protocols have removed resting fibroblasts. Instead, they have focused on removal of activated fibroblasts and only examine deletion efficiency within this subset. While the cutoff we used was 45%, the average deletion was ~70% (69.9% +/- 13.9). In the previous manuscript, we also did not perform any injury. Nonetheless, we can appreciate the reviewer’s concerns and have removed the range of deletion statements throughout the manuscript.

One important and surprising finding in this paper is that the cells surviving the ablation did not expand to make up for the lost numbers. However, I was surprised to see that most of the evaluation of the extent of depletion over time relied on the expression of PDGFRa itself, a gene whose expression has been reported to be modulated, that is expressed at different levels in different subsets (based on published scRNAseq data) and that is expressed by a single allele in this experimental setup. In some cases, the results are expressed a "stained area/tissue area", when clearly it would have been more appropriate to provide cell counts. At least, they should provide FACS plots showing the gating used for quantification to reassure the readers that PDGFRa was expressed at comparable levels in mice of different ages. Given that it is now widely accepted that cardiac fibroblasts are heterogenous in many ways, it would be important to understand whether this ablation affects this heterogeneity. In other words, does ablation preferentially remove subpopulations of fibroblasts (such as more proliferative cells that express higher levels of Pdgfra) more than others? Knowledge of this may explain possible reasons for the lack of replenishment of fibroblasts. I think it would be worth using scRNAseq in this context.

We have addressed most of these concerns above with the exception of performing single cell sequencing. We hope that the primary cell isolation and use of Col1a1GFP help to alleviate the heterogeneity concerns raised by this reviewer.

The CRE strain used is active in a number of other organs and anatomical locations, ranging from lung fibroblast to adipogenic cells in fat depots to oligodendrocyte progenitors in the CNS. Did the authors look into these organs, or notice any systemic effects of their depletion protocol?

We now include a blood metabolic panel. While we appreciate that this does not reflect other organ function such as the liver, spleen, skin etc. We hope that it provides an idea of the general health of the animals. Supplemental table 2. For the reviewer’s information, we do observe reduced fibroblasts in the lung, spleen, kidney, and liver. The lung is complicated by the fact that there are other fibroblasts that never express PDGFRa. We have not examined skin or cornea (both of which would also be predicted to have a reduction). Surprisingly, oligodendrocytes do not seem to be perturbed by PDGFRa ablation. Potentially because they either express too low of a level to induce DTA or they are repopulated by some progenitor cell. We have not explored the “other” fibroblast population numbers in these organs.

Based on the data presented, the hearts contained somewhere between half and one third of their normal number of fibroblasts, suggesting that one or two cell divisions would be sufficient to bring these cells back to normal levels. Following an MI, these cells enter rapid proliferation, and their numbers increase significantly. What happens in the ablated heart to fibroblast numbers? Do they increase as expected, do they plateau at lower or similar levels to the ones observed in controls? Does the scar still contain the same number of collagen-expressing cells?A time resolved analysis of PDGFRa cells in the context of the two damage models would be very informative and possibly hint at mechanism, and I was surprised to see the analysis limited to one time point and performed by quantifying genes rather than cells themselves.

These concerns are addressed above.

The proteomic analysis of the de-cellularized heart is a strong point of the paper. However, some of the changes observed may not be a direct effect of fibroblast ablation. For example, one of the proteins the authors reported to be upregulated in the fibroblast-ablated hearts is NG2, which is not expressed to any significant extent in fibroblasts. Rather, it is expressed at high levels in mural cells such as pericytes and vascular smooth muscle cells. Does the increase in NG2 indicate an upregulation of the protein in mural cells or increase (maybe compensatory) in the numbers of mural cells? Adding mural cells to the FACS-based quantification of inflammatory, endothelial and fibroblastic cells would help in answering this.

These concerns are addressed above.

I am a bit confused by the fact that the authors, although reported decreased LV internal diameter, found that cardiomyocytes did not increase in size. Usually, the reason for decreased chamber size would be hypertrophy, especially if it is observed in systole and diastole. How do the authors explain this?

Our data indicate that shortening and Ca kinetics are slowed, while shortening amplitude is decreased. Thus, the slowed kinetics for both shortening and Ca match, with the slowed Ca decline most likely leading to slowed relaxation. While Ca transient amplitude is normal, shortening is decreased. These data would suggest that there is a decrease in Ca sensitivity. In our ablated hearts, the basement membrane is disrupted that could decrease myofilament calcium sensitivity (PMID: 28018228) through various mechanisms such as changes in myofilament phosphorylation (PMID: 34641704).

I am surprised that the authors did not attempt to measure diastolic function in the fibroblast-ablated hearts following damage in either the MI or the Ang II/phenylephrine models. The parameters they report are more relevant to systolic function (ejection fraction) and chamber dimension (LVID). However, I would have expected some measurement of transmitral flow such as E/A ratio or E wave deceleration time from a 4- chamber apical view, or even IVRT. This is important as ECM/fibrotic changes are expected to affect LV compliance/stiffness (i.e. diastolic function) usually before systolic involvement.

Addressed above.

How was the collagen area measured in the 2 damaged models? By antibody staining or using Trichrome staining? Why in the authors' opinion did fibroblast ablation decrease collagen area in the Ang II/phenylephrine model but not the MI model? Again these questions may be answered by analyzing the proliferative response to these insults.

Addressed above.

A decrease in shortening without a decrease in calcium transient amplitude, as seen in the ablated hearts, indicates calcium desensitization. An important pathway that regulates calcium sensitivity and the actin cytoskeleton at the same time is the RhoA/Rho kinase pathway. Did the authors find any changes in expression of the proteins in this pathway?

Addressed above.

Reviewer #3 (Recommendations for the authors):In this manuscript, Kuwabara and colleagues use genetic ablation to reduce the number of fibroblasts resident to the heart. At baseline, the authors observe that fibroblast numbers stay proportionally low after ablation, but with very minimal effects to the structure or composition of the extracellular matrix. Fibroblast ablation prior to myocardial infarction is shown to be beneficial to cardiac function without affecting relative abundance of scar tissue, whereas in an Ang/PE model of fibrosis collagen deposition is impaired and systolic function is preserved.Major Concerns1. For the myocardial infarction model presented in Figure 4, it is fascinating that no rupture is observed and that the relative collagen fraction between the ablated mice and controls is largely similar by 10 days. (A) Have the non-ablated fibroblasts proliferated and normalized the fibroblast numbers back to wildtype levels? (B) Is it possible that collagen area as a percentage of tissue area may not be the most informative metric of fibrosis due to the significantly altered cardiac dimensions in mice with ablated fibroblasts? (C) It is puzzling that fibroblast ablation has so little impact on the ECM opening up questions regarding the cell state heterogeneity of the non ablated fibroblasts populations- are they all behaving like activated fibroblasts or matrifibrocytes to maintain the matrix?

Addressed above.

2. This study paid a lot of attention to mechanisms of ECM deposition and remodeling but very little attention to degradation and turnover of the matrix. Are these mechanisms inhibited such that the matrix is no longer turning over once fibroblasts are ablated?

Addressed above.

3. What is the basis for the reduction in collagen deposition observed for the Ang/PE model but not for the infarction model?

Addressed above.

4. If the TAILS proteomics data suggests that cardiomyocyte remodeling may be occurring, is this finding reflected in the geometry of isolated myocytes at baseline measured for their contraction characteristics in figure 7?

Addressed above.